# Biomass burning aerosols in most climate models are too absorbing

Hunter Brown [1], Xiaohong Liu[1,2 ✉], Rudra Pokhrel [1,3], Shane Murphy [1], Zheng Lu[1,2], Rawad Saleh[4],
Tero Mielonen[5], Harri Kokkola [5], Tommi Bergman [6], Gunnar Myhre [7], Ragnhild B. Skeie [7],
Duncan Watson-Paris [8], Philip Stier [8], Ben Johnson[9], Nicolas Bellouin [10], Michael Schulz [11],
Ville Vakkari[12,13], Johan Paul Beukes[13], Pieter Gideon van Zyl[13], Shang Liu[14] & Duli Chand[15]

Uncertainty in the representation of biomass burning (BB) aerosol composition and optical properties in climate models contributes to a range in modeled aerosol effects on incoming solar radiation. Depending on the model, the top-of-the-atmosphere BB aerosol effect can range from cooling to warming. By relating aerosol absorption relative to extinction and carbonaceous aerosol composition from 12 observational datasets to nine state-of-the-art Earth system models/chemical transport models, we identify varying degrees of over-estimation in BB aerosol absorptivity by these models. Modifications to BB aerosol refractive index, size, and mixing state improve the Community Atmosphere Model version 5 (CAM5) agreement with observations, leading to a global change in BB direct radiative effect of $-0.07\ \mathrm{W\,m^{-2}}$, and regional changes of $-2\ \mathrm{W\,m^{-2}}$ (Africa) and $-0.5\ \mathrm{W\,m^{-2}}$ (South America/Temperate). Our findings suggest that current modeled BB contributes less to warming than previously thought, largely due to treatments of aerosol mixing state.

[1] Department of Atmospheric Science, University of Wyoming, Laramie, WY, USA. [2] Department of Atmospheric Sciences, Texas A&M University, College Station, TX, USA. [3] Department of Physics, North Carolina A&T State University, Greensboro, NC, USA. [4] Air Quality and Climate Research Laboratory, University of Georgia, Athens, GA, USA. [5] Finnish Meteorological Institute, FI-70211 Kuopio, Finland. [6] Climate System Research, Finnish Meteorological Institute, FI-00101 Helsinki, Finland. [7] Center for International Climate and Environmental Research – Oslo (CICERO), Oslo, Norway. [8] Atmospheric, Oceanic and Planetary Physics, Department of Physics, University of Oxford, Oxford, UK. [9] Met Office, Exeter, UK. [10] Department of Meteorology, University of Reading, Reading, UK. [11] Norwegian Meteorological Institute, Oslo, Norway. [12] Finnish Meteorological Institute, FI-00101 Helsinki, Finland. [13] Atmospheric Chemistry Research Group, Chemical Resource Beneficiation, North-West University, Potchefstroom, South Africa. [14] School of Earth and Space Sciences, University of Science and Technology of China, Hefei, China. [15] Atmospheric Sciences and Global Change Division, Pacific Northwest National Laboratory, Richland, WA, USA. ✉email: xiaohong.liu@tamu.edu

Large uncertainties in the representation of biomass burning (BB) aerosols in Earth system models (ESM) and chemical transport models (CTM)[1,2] increase the range in their simulated climate impact[3]. Reducing this range through improvements of aerosol emissions, atmospheric processes, and microphysical and optical properties can elucidate the effect of BB aerosols on climate[4–6], human health[7,8], as well as their role in the carbon cycle[9].

Biomass burning aerosol make up a majority of primary combustion aerosol emissions[1,10], with the main sources of global BB mass being Africa (~52%), South America (~15%), Equatorial Asia (~10%), Boreal forests (~9%), and Australia (~7%)[11]. The composition, size, and mixing state of BB aerosols determine the optical properties of smoke plumes in the atmosphere, which in turn is a major factor in dictating how they perturb the energy balance in the earth system. Aerosols affect the top of the atmosphere (TOA) radiative flux by scattering and absorbing incoming shortwave radiation (i.e., aerosol-radiation interactions) and by modifying cloud droplet number concentrations and other cloud properties (i.e., aerosol-cloud interactions)[2,12,13]. While all aerosols scatter light – reducing the surface energy flux – only a few species are significant absorbers of incoming solar radiation and increase the energy stored in the earth system. These absorbing aerosol species include BC, dust, and absorbing organic aerosol. The latter of these species, often called brown carbon (BrC)[14], is a recent addition to CTMs and ESMs[15–18]. Condensable gas-phase species create a secondary aerosol coating of predominantly scattering material that either fully or partially coats these absorbing aerosol[19]. In the case of BC, secondary and primary coatings can enhance absorption of light[20–23].

The International Panel on Climate Change Assessment Report 5 (IPCC AR5) estimates the radiative forcing of BB aerosols due to aerosol-radiation interactions (RF$_{ari}$) to be 0.0 (±0.2) W m$^{-2}$ based on 12 ESMs/CTMs from the Aerosol Comparison (Aero-Com) Phase II experiment[2,24]. RF$_{ari}$ is the difference in TOA radiative flux between present day BB (2011) and preindustrial BB (1750) conditions. This near-zero BB RF$_{ari}$ reflects the balance between the strongly absorbing BC and the more scattering organic and sulfate aerosols that make up model BB smoke. The range in forcing reflects uncertainties in model BB emissions[1,25], BC vertical profiles[26], BC and POA aerosol optical properties in BB smoke[15,27–29], BB aerosol size distributions[30,31], and aerosol mixing state[16,23,31–34]. Furthermore, specific cases of modeled aerosol radiative forcing above marine stratocumulus clouds depend on a combination of model treatments that include BC optical properties, aerosol injection height, aerosol vertical transport, and the parameterization of cloud top albedo[35,36]. This study examines the sensitivity of aerosol radiative effects to three of these model parameters: optical properties, size distributions, and mixing state of BB aerosols.

We approach the validation of these aerosol microphysical and optical properties through BB observations of single scattering albedo (SSA) versus BC-to-total-carbon (BC + OC) ratio (BC:TC) from aircraft campaigns, ground-based sites, and laboratory BB experiments (note: OC represents the mass of carbon associated with OA concentrations, the latter of which also has mass contributions from hydrogen, oxygen, and nitrogen). Validations of modeled BB optical properties typically rely on measurements of bulk SSA and aerosol optical depth (AOD) collected with satellites and ground-based sun-photometer sites (Aerosol Robotic Network (AERONET); http://aeronet.gsfc.nasa.gov/)[1,6,15,35,37,38]. While both of these datasets tend to have better global coverage than more intermittent aircraft field campaigns, a benefit of using in situ observations in this comparison is that these data can provide information on an aerosol species basis and do not suffer from the uncertainties inherent in the bulk observations of satellite and

ground-based photometer remote sensing data. Satellite uncertainties can stem from cloud masking techniques, surface reflectance estimation, and estimations of aerosol type[39], while AERONET uncertainties derive from cloud masking techniques, AOD sampling thresholds, and corrections at different sampling angles[40]. One challenge that arises from comparing global models to in situ observational datasets is that observations describe conditions at finer temporal and spatial resolutions than are typically resolved by ESMs and CTMs. By choosing the BB variables SSA and the BC:TC, systematic biases in simulated BB optical and compositional properties can be identified that are less dependent on sample resolution and temporal variation in BB aerosol emissions (though representation errors in the comparison may still exist[41]). This framework is used to compare treatments of BB aerosol from nine state-of-the-art ESMs and CTMs. Sensitivity tests are conducted with the NCAR Community Atmosphere Model version 5 (CAM5)[42] to analyze the global radiative forcing consequences of improvements within this framework, with implications for the other models in this study.

## Results

**Biomass burning aerosol properties affecting SSA.** For the model evaluation, this study presents a comprehensive set of BB observations. This data comes from 12 field and laboratory campaigns conducted in a number of source regions (e.g., North America, South America, Africa, India, Indonesia; Fig. 1, Table 1), with the criterion that BB carbonaceous aerosols dominate each data point (Methods). The BB aerosols range in age from minutes to a few days (Supplementary Table 4). Constraining the contribution of carbonaceous aerosols minimizes the optical influence from inorganics measured by aerosol mass spectrometry (i.e., ammonium, sulfate, chloride, nitrate). This also minimizes the influence of aged aerosols, which may have larger mass contributions from sulfate and non-BB secondary organic aerosol (SOA). Gaps in the dataset coverage are largely due to the scarcity of global BB samples. Instrument uncertainty estimates for these data are represented by error bars in Fig. 1 (see Methods).

Most of the model data used in this study originated from the Biomass Burning Emissions Experiment of the AeroCom (Aerosol Comparison between Observations and Models; http://aerocom.met.no) Phase III. These models include the Community Atmosphere Model version 5.3[43] (CAM5.3), Hadley Center Global Environment Model version 3[38,44] (HadGEM3), two versions of the European Center Hamburg Atmospheric Model version 6.3 (ECHAM6.3-HAM-SALSA2[45,46] and ECHAM6.3-HAM2.3-M7[47,48]—ECHAM6.3-SALSA2 and ECHAM6.3-HAM2.3, respectively), and the Oslo CTM version 2[49,50] (OsloCTM2). We utilize the control and no-BB simulations from these models, with BB emissions from the Global Fire Emissions Dataset, version 3 (GFEDv3)[51], and the simulation year 2008. A one-year simulation (2005) from the Goddard Earth Observing System CTM[52] (GEOS-Chem) is used from Saleh et al.[16]. Lastly, we include three versions of CAM: a default version (CAM5.4) that we use for sensitivity studies with differing aerosol microphysics and radiative properties in the model (Supplementary Tables 1,2), as well as two simulations including BrC from Brown et al.[17] (BrC with and without photochemical bleaching—CAM5.4_BrCbl and CAM5.4_BrC, respectively). Similar to observations, model data is limited to BB regions by isolating grid cells dominated by BC and OA from BB sources (Methods).

Figure 1 shows SSA (at 550 and 700 nm) versus BC:TC in BB regions from observations, while Fig. 2 shows the same data in Fig. 1a overlaid by model predictions. Both the observations and the model data in Fig. 2 show a decreasing linear relationship

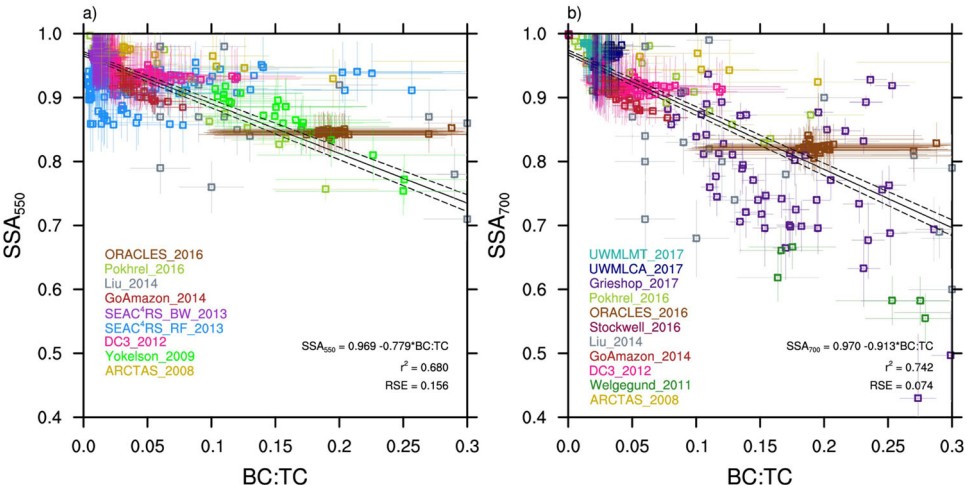

**Fig. 1 Observations of biomass burning single scattering albedo versus black carbon to total carbon ratio. a** Single scattering albedo (SSA) at 550 nm wavelength versus black carbon to total carbon ratio (BC:TC) and **b** SSA at 700 nm wavelength versus BC:TC. Included are the linear regression equation, the $r^2$ value, and the residual standard error (RSE). Dotted lines represent the 95% confidence intervals of the fit. Different observations are color-coded, and measurement uncertainty is plotted when available.

between SSA and BC:TC, with slopes of $-0.779 \pm 0.017$ (observations) and $-0.625$ to $-2.17$ (model range) (Supplementary Table 9). Interannual variation in slope is included for CAM5.4, CAM5.4_BrC, and CAM5.4_BrCbl (Fig. 2a, b), and depends on the global distribution of BB emissions (Supplementary Fig. 11). Underestimation of SSA in these models compared to observations is consistent year-to-year. Spread in model SSA at a given BC:TC is related to aerosol water content in the grid cells, with larger SSA, and much of the North America (NAm) and Northern Asia (NAs) grid cells, related to higher aerosol water content (Supplementary Fig. 12). For ECHAM6.3-HAM2.3 and ECHAM6.3-SALSA2, some of the points with lower SSA/higher BC:TC correlate with upper level grid cells in the model (Supplementary Fig. 14). This is partly attributed to the emission of hydrophobic aerosols in these two models[46,47], which may be more prone to vertical transport due to reduced wet scavenging. Modeled BB emission heights could also play a role in this vertical spread in SSA, as emissions that escape the planetary boundary layer (PBL) are more likely to have longer lifetimes and undergo long-range transport.[53]

Global variation in BB emissions can be seen in the regional dependence of BC:TC and SSA in Fig. 2. In the models, NAm and NAs are characterized by lower BC:TC and higher SSA. On the other hand, Africa (Afr), South America (SAm), and South Eastern Asia (SEas) are characterized by higher BC:TC and lower SSA. This results from GFEDv3 used in the simulations, which has higher BC:TC in regions classified as savannah/grasslands/woodlands/peat than boreal forests[11]. It is worth noting that the GFEDv3 emission data set assumes higher BC emission factors for peat than the more updated GFED version 4 (GFEDv4)[25], the latter of which agrees better with observations of smoldering peat combustion[54,55]. This can explain the high BC:TC in many of the models from the SEas region compared to the observational mean from Stockwell et al.[55] (BC:TC = 0.0004). The occurrence of grassland/savannah fires in NAm and NAs contribute to lower SSA/higher BC:TC in these regions, with a larger percentage of boreal forest fires occurring in NAm than in NAs.

Regionally dependent BC:TC is also visible in the observational data, with regional means denoted at the bottom of each panel of Fig. 2. Overall, model simulations do not reproduce the spread in the BC:TC of the observations, overestimating NAm/NAs BC:TC and underestimating Afr BC:TC. Regionally, the higher model

BC:TC ratio in the NAm and NAs regions could be due to an overestimation of modeled BC in the Arctic[26] and/or an underestimation in global OA[56]. This may be partially explained by lack of SOA formation in the models, as the low BC:TC (< 0.05) Afr, SAm, and SEas model data from CAM5.3 (Fig. 2c) are examples of grid cells influenced by SOA formation on BB aerosol (Supplementary Fig.15). Lower observed BC:TC in SAm observations compared to most modeled SAm grid cells may be due to higher observed SOA formation rates[57] than are simulated in the models. These differences between model and observations could also come from the model emission datasets. The emission ratios in these models are ultimately decided by GFEDv3, which reports average elemental carbon (EC; assumed to be BC in climate models) to OC emission ratios based on different fuel/land cover types[58]. This may explain the lack of low and high BC:TC, where the more extreme cases have been averaged out. Another possible explanation for overestimated BC:TC in the models may be the thermal-optical analysis technique used to determine EC in the emission inventories. This technique assumes that OA is non-absorbing in the visible spectrum, which could result in erroneous classification of BrC as EC and lead to emissions of EC that are biased high[18].

Differences in SSA model behavior in Fig. 2 can be attributed to unique model treatments of BB/BC refractive index (RI), mixing state, and BB aerosol size (Supplementary Table 2). HadGEM3 (Fig. 2f) and OsloCTM2 (Fig. 2g) both have overall better agreement with observational SSA compared to most other models due to a smaller imaginary part (absorption/attenuation term) in their BC RI (HadGEM3) and observationally constrained BB RI (OsloCTM2). Every model in this comparison treats aerosol species as internally mixed (well mixed or core–shell[59]) within aerosol modes (CAM5.3, CAM5.4, ECHAM6.3-HAM2.3, HadGEM3, OsloCTM2), size classes (ECHAM6.3-SALSA2), or a bulk aerosol representation (GEOS-Chem). The exceptions to this are the upper two GEOS-Chem simulations in Fig. 2h, which treat aerosols as externally mixed. One of these simulations treats BB OA as non-absorbing and has the best agreement with observations of all of the models addressed in this study. However, this should not be taken as an indication that assuming BB OA to be non-absorbing is the most physically sound assumption. Scattering efficiency, thus SSA, is strongly dependent on aerosol size. It is challenging to decouple the effects of aerosol

**Table 1 Observational datasets used in this study along with their locations and sample collection dates. Single dates indicate that data was used from a single day of the campaign.**

| Acronym | Data source | Collection date (s) | Source region | Reference |
|---|---|---|---|---|
| ARCTAS_2008 | Arctic Research of the Composition of the Troposphere from Aircraft and Satellites (ARCTAS) | 2008/07/01 | Saskatchewan, Canada | Jacob et al.[100] |
| DC3_2012 | Deep Convective Clouds and Chemistry (DC3) | 2012/06/22 | Colorado, USA | Barth et al.[101] |
| GoAmazon_2014 | Green Ocean Amazon (GoAmazon) | 2014/09/30-10/01 | Manaus, Brazil | Martin et al.[102] |
| Grieshop_2017 | Grieshop et al.[103] | 2011/09 – 2012/08 | Karnataka State, India | Grieshop et al.[103] |
| Liu_2014 | Fire Laboratory at Missoula Experiment 4 (FLAME-4) | 2012/10-11 | Global | Liu et al.[95] |
| ORACLES_2016 | Observations Above Clouds and their Interactions (ORACLES) | 2016/09/02 | Southern Africa | Zuidema et al.[104] |
| Pokhrel_2016 | Fire Laboratory at Missoula Experiment 4 (FLAME-4) | 2012/11/15-16 | Global | Pokhrel et al.[94] |
| SEAC⁴RS_2013 | Studies of Emissions and Atmospheric Composition, Clouds and Climate coupling by Regional Surveys (SEAC⁴RS) | | | Toon et al.[105] |
| SEAC⁴RS_BW_2013 | Big Windy Fire | 2013/08/06 | Oregon, USA | |
| SEAC⁴RS_RF_2013 | Rim Fire | 2013/08/26 | California, USA | |
| Stockwell_2016 | Stockwell et al.[55] | 2015/09/05-06 | Central Kalimantan, Indonesia | Stockwell et al.[55] |
| UWML_2017 | University of Wyoming Mobile Lab | | | Foster et al.[93] |
| UWMLCA_2017 | | 2017/10/13-17 | California, USA | |
| UWMLMT_2017 | | 2017/08/27-29 | Montana, USA | |
| Welgegund_2011 | Vakkari et al.[66]Vakkari et al.[97] | 2010/09/01-2011/08/16 | North West Province, South Africa | Vakkari et al.[66]Vakkari et al.[97] |
| Yokelson_2009 | Megacity Initiative Local and Global Research Observations (MILAGRO) | 2006/03/23 | Yucatan Peninsula, Mexico | Yokelson et al.[65] |

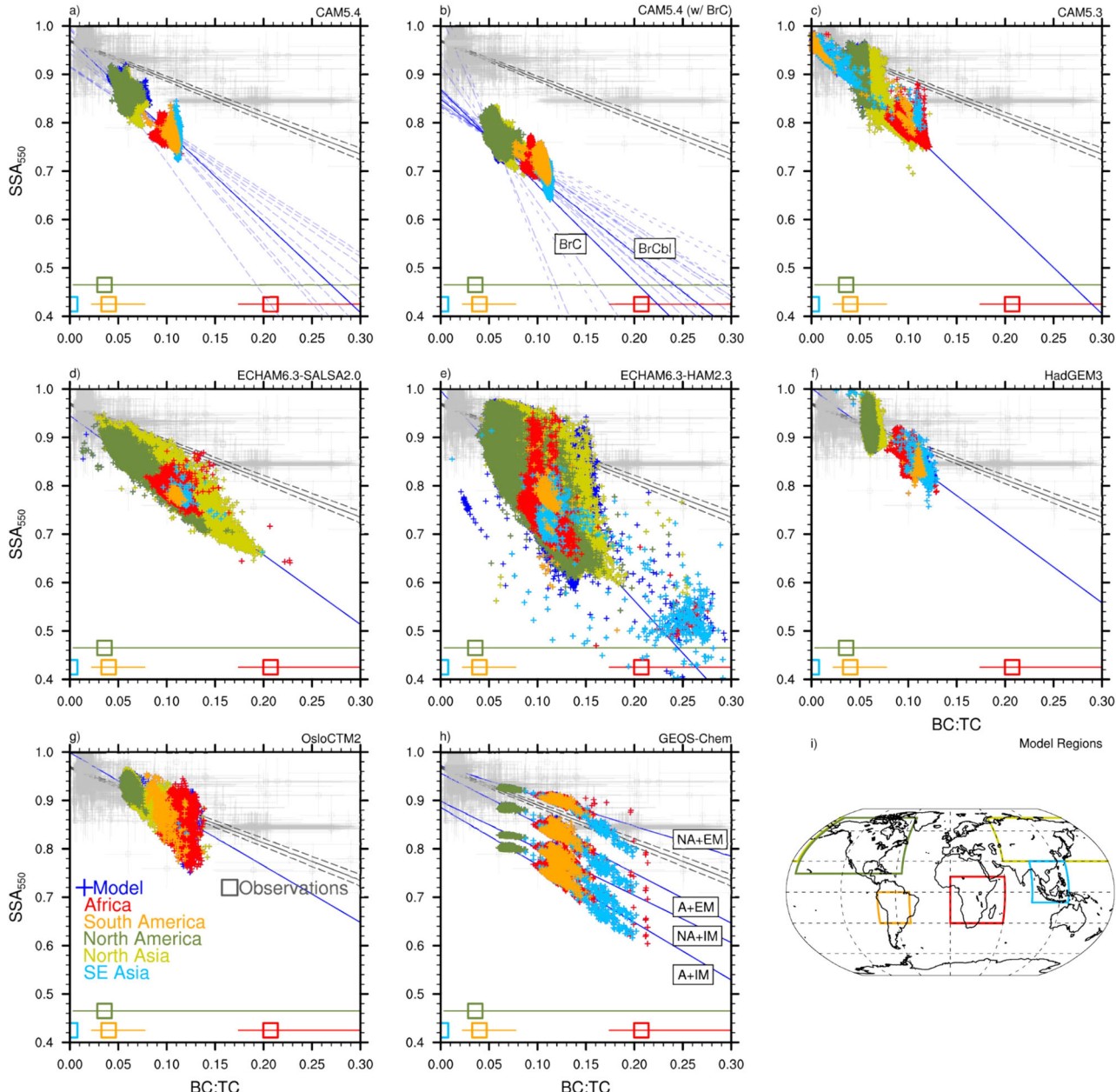

**Fig. 2 Comparison of observed and modeled biomass burning single scattering albedo versus black carbon to total carbon ratio.** All panels plot biomass burning (BB) single scattering albedo (SSA) at 550 nm wavelength versus black carbon to total carbon ratio (BC:TC). Observational data is in gray. The model data is from **a** this study, **b** Brown et al.[17], **c–g** AeroCom Phase-III simulations, and **h** Saleh et al.[16]. They are **a** CAM5.4, **b** CAM5.4 (w/ brown carbon (BrC)), **c** CAM5.3, **d** ECHAM6.3-SALSA2, **e** ECHAM6.3-HAM2.3, **f** HadGEM3, **g** OsloCTM2, and **h** GEOS-Chem. Model data is representative of BB influenced regions which are color-coded in the model output and are specified on the global plot in **i**, defined as Africa (red), South America (orange), North America (green), Northern Asia (yellow), and Southeastern Asia (light blue). Observational data from the specific regions are averaged, and the regional average and range of the BC:TC observational data is included at the bottom of the plot (regionally color-coded squares). The two CAM5.4 simulations with BrC **b** represent brown carbon with and without photochemical bleaching (BrC and BrCbl, respectively). The four GEOS-Chem simulations **h** represent the four model simulations from Saleh et al.[16]: no BrC and externally mixed aerosols (NA + EM); BrC and externally mixed aerosols (A + EM); no BrC and internally mixed aerosols (NA + IM); and BrC and internally mixed aerosols (A + IM). Interannual variation in model slope and intercept is represented by dashed blue lines in **a** and **b** (in **b**, long dash = BrC, short dash = BrCbl). The solid gray line is the best fit to observations, with dashed lines representing the 95% confidence intervals of the fit. The best fit to the model data is represented by the solid blue line.

size treatment and BB OA light-absorption properties on model differences in SSA. However, based on Mie calculations in the size range of interest (diameters from 100 to 400 nm), it is known that larger aerosols scatter more visible light, increasing BB aerosol SSA. It is thus possible that the small SSA values in the models relative to observations are due to underestimation of BB aerosol

sizes. Figure 3 shows a comparison of the modeled and observed mass absorption cross-section at 550 nm ($MAC_{BB}$; m² g⁻¹) plotted against BC:TC. $MAC_{BB}$ is significantly less dependent on aerosol size than SSA[60]. Some models in Fig. 3 show a deviation from observations at higher BC:TC (Fig. 3a–d), indicating an overestimation in BB aerosol absorption. Models that show less

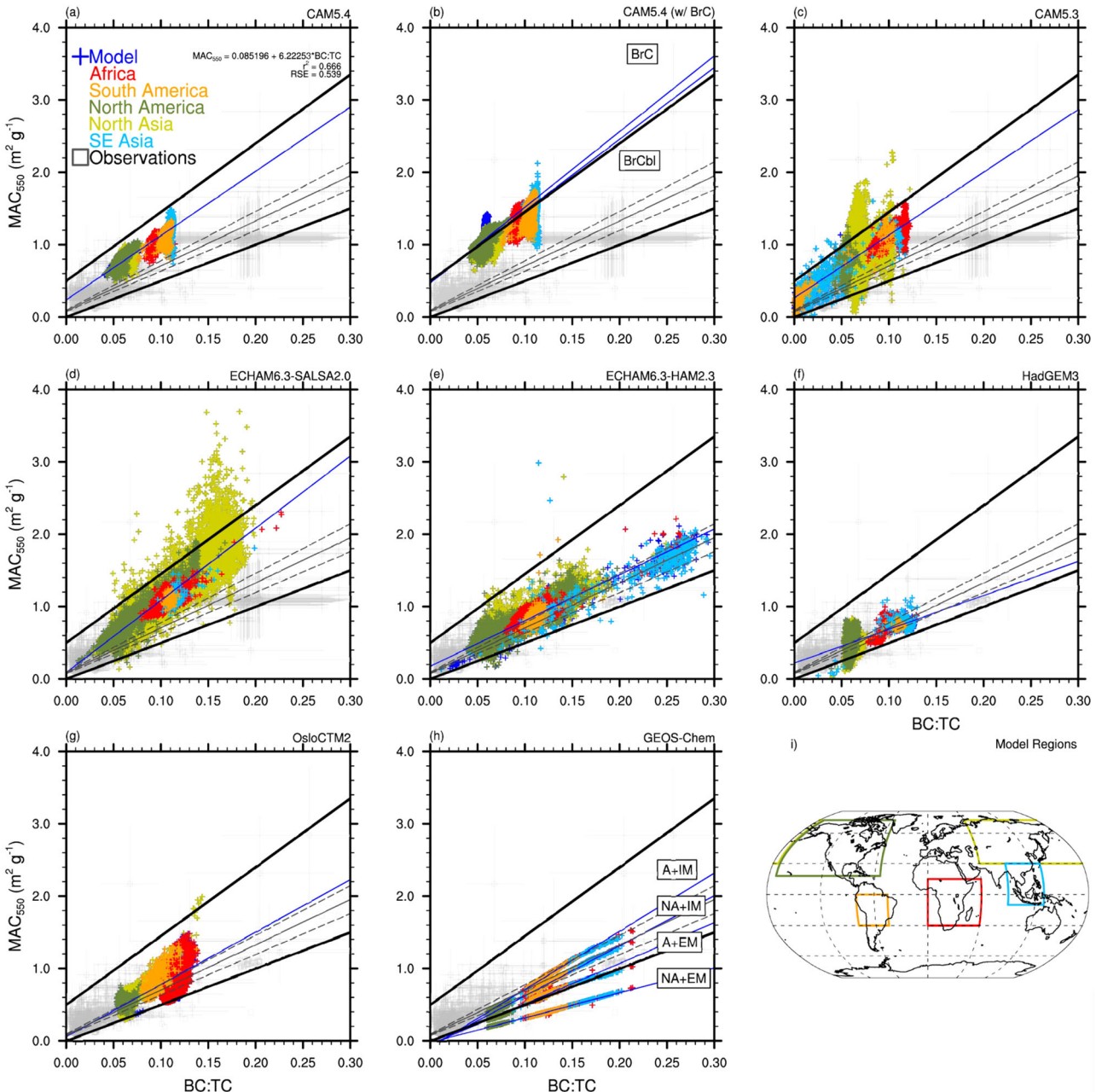

**Fig. 3 Comparison of observed and modeled biomass burning mass absorption cross-section versus black carbon to total carbon ratio.** Biomass burning (BB) mass absorption cross-section (MAC) depends on the 550 nm absorption coefficient ($\beta_a$), as well as BB black carbon and organic aerosol concentrations ([BC], [OA]) (MAC = $\beta_a$ /([BC] + [OA]; $m^2 g^{-1}$). All panels plot BB MAC versus black carbon to total carbon ratio (BC:TC). Observations are in gray. The model data is from **a** this study, **b** Brown et al.[17], **c–g** AeroCom Phase-III simulations, and **h** Saleh et al.[16]. They are **a** CAM5.4, **b** CAM5.4 (w/ brown carbon (BrC)), **c** CAM5.3, **d** ECHAM6.3-SALSA2, **e** ECHAM6.3-HAM2.3, **f** HadGEM3, **g** OsloCTM2, and **h** GEOS-Chem. Model data is representative of BB influenced regions which are color-coded in the model output and are specified on the global plot in **i**, defined as Africa (red), South America (orange), North America (green), Northern Asia (yellow), and Southeastern Asia (light blue). The two CAM5.4 simulations with BrC **b** represent brown carbon with and without photochemical bleaching (BrC and BrCbl, respectively). The four GEOS-Chem simulations **h** represent the four model simulations from Saleh et al.[16]: no BrC and externally mixed aerosols (NA + EM); BrC and externally mixed aerosols (A + EM); no BrC and internally mixed aerosols (NA + IM); and BrC and internally mixed aerosols (A + IM). The solid gray line is the best fit to observations, with dashed lines representing the 95% confidence intervals of the fit. Thick black lines constrain a reasonable range in BB MAC[18]. The best-fit to the model data is represented by the solid blue line.

divergence from observations (Fig. 3e–h) indicate a disagreement that is due in part to size dependent scattering differences between the model and observations. Most of the model data fits within the envelope of reasonable MAC$_{BB}$[18], with outliers in Fig. 3 likely due to some combination of overestimated BC

absorption enhancement or too high an imaginary part of the BB RI due to BC and/or BrC.

What this collection of models shows is that all of the models overestimate BB aerosol absorption relative to extinction to some extent. Some models perform better than others in this

comparison, especially with smaller BB aerosol imaginary refractive indices and external mixing assumptions. Of the models included in this comparison, two CAM5.4 simulations and two GEOS-Chem simulations include the effects of BrC on aerosol absorption. CAM5.4 simulations with BrC from Brown et al.[17] exacerbate the excessive absorption in the model, with photochemical bleaching effects (CAM5.4_BrCbl) slightly improving model-observation slope agreement. At shorter wavelengths (400 nm) where the effects of BrC absorption are greater, CAM5.4 agrees better with observations (Supplementary Fig. 16) than CAM5.4_BrC (Supplementary Fig. 17), which continues to exhibit lower SSA than observations. This comparison emphasizes that CAM5.4 SSA is too low at both 400 and 550 nm wavelengths, and including BrC in the model further reduces model SSA. In order to help understand the underlying causes of the BB absorption enhancement in the models, the following section will address areas of improvement in CAM5.4, and how these improvements affect BB SSA.

**Adjustment to CAM5.4 and different model treatments**. The factors affecting the calculation of aerosol SSA are aerosol size, composition, and mixing state. In CAM5.4, aerosols are treated in four different size modes with the Modal Aerosol Module (MAM4)[61]. Biomass burning aerosols are treated as both freshly emitted (Mode 4) and aged (Mode 1). Due to the prevalence of Mode 4 aerosols in BB grid cells isolated in Fig. 2a, and their similarity in lifetime to observed BB plumes, they will be the focus of the BB modifications (Methods).

The default simulation (CAM5.4) is plotted in Fig. 2a. The following modifications to this base version are made in separate simulations (simulation names in parenthesis):

**BC refractive index** (CAM_BCRI): The BC refractive index used in CAM5.4 (1.95–0.79i) is the most absorbing (i.e., lowest void fraction) of the recommended, observationally inferred BC refractive indices in Table 5 from Bond and Bergstrom[27] (Supplementary Table 2). Given the better performance of models with less absorbing, observationally derived imaginary refractive indices (HadGEM3, OsloCTM2), the lowest recommended BC RI value from Bond and Bergstrom[27] (1.75–0.63i) was implemented for the simulation CAM_BCRI.

**Aerosol size** (CAM_DG160): The globally averaged aerosol size in CAM5.4 is smaller than available observations of BB aerosol size distributions (Supplementary Fig. 3). In CAM5.4, Mode 4 mean aerosol diameter is constrained to the range 10–100 nm. This was modified in CAM_DG160 to allow a maximum mean diameter of 300 nm. The actual aerosol size in the model varies based on the bulk aerosol density along with number and mass emissions (Methods, Eqs. 8, 9)[43]. In CAM_DG160, the number emission for Mode 4 was reduced —leaving mass emissions unchanged—in order to increase the global, BB, Mode 4 number mode diameter to better match the average observational number mode diameter of 160.1 nm (to within 1%) (Supplementary Tables 7,8).

**Aerosol Mixing State** (CAM_EMIX): Mode 4 BB aerosols are treated as externally mixed POA and BC. This is based on evidence that idealized internal mixing assumptions used in climate models have been shown to overestimate BC absorption enhancement when compared to observations[62] and lead to greater BC absorption enhancement than non-uniform internal mixtures based on observed mixed aerosol[19]. Additionally, GEOS-Chem simulations in Fig. 2f that assume externally mixed aerosols agree better with observational SSA. Accumulation mode (Mode 1) aerosols remain well-mixed internally (default, Supplementary Table 2) along with the Mode 4 anthropogenic aerosol, only treating the freshly

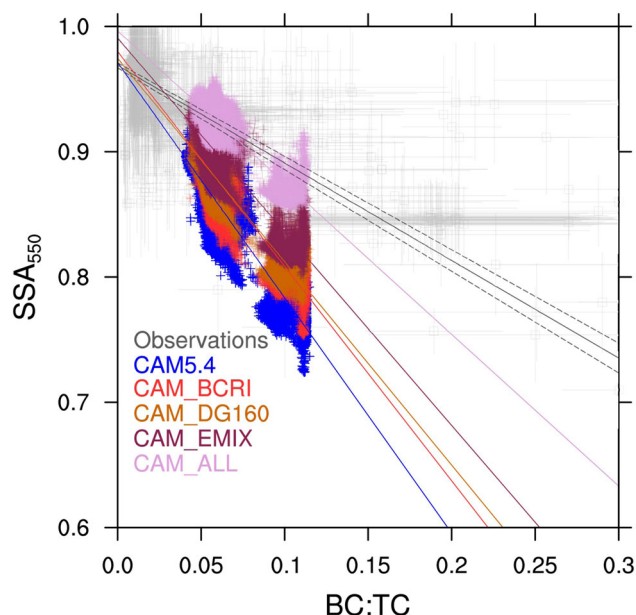

**Fig. 4 Model improvement compared to observations with different biomass burning microphysical and radiative properties.** Comparison of observed biomass burning (BB) single scattering albedo (SSA) at 550 nm wavelength versus black carbon to total carbon ratio (BC:TC) to CAM5.4 sensitivity tests. Observations are in gray. The CAM5.4 sensitivity tests are for different BB aerosol treatments: default CAM5.4 (CAM5.4; blue), CAM5.4 with decreased BB black carbon refractive index (CAM_BCRI; red), CAM5.4 with increased BB aerosol size (CAM_Dg160; gold), CAM5.4 with externally mixed, fresh BB aerosol (CAM_EMIX; maroon), and CAM5.4 with all of the previous changes (CAM_ALL; pink).

emitted BB aerosol as externally mixed. This treatment is for radiative calculation purposes only; dry and wet deposition—in addition to cloud interactions—remain the same as in CAM5.4.

**Combined** (CAM_ALL): All of the aforementioned modifications are applied to CAM5.4.

When these changes are made in concert to the CAM5.4 model, all of the modifications improve the model agreement with observations (Fig. 4). Model SSA improvement is also noted in comparisons with AERONET BB sites (Supplementary Fig. 20). CAM_BCRI and CAM_DG160 have the same change in globally averaged SSA (+2.1%), while CAM_EMIX has more than twice the change in SSA of these simulations (+4.6%). CAM_ALL has the greatest improvement of any of the CAM5.4 sensitivity tests, increasing the globally averaged SSA by 7.4%. When looking at the change in BB aerosol absorption relative to extinction (1-SSA), the CAM_BCRI, CAM_DG160, CAM_EMIX, and CAM_ALL simulations have a change of −12.8%, −13.5%, −28.4%, and −45.4%, respectively (Supplementary Tables 10,11). As this study avoids changes in mass emissions and emission factors, there is little change in the BC:TC ratios between the sensitivity tests.

Modifications to BB radiative properties in CAM5.4 yield improvement in the SSA versus BC:TC framework. Next, the impact of these modifications on climate projections will be addressed by assessing the radiative impacts of these modifications in CAM5.4.

**Radiative effect of microphysical modifications**. The IPCC AR5 report cites the difference in BB radiative effects between present day and preindustrial simulations to determine the BB radiative

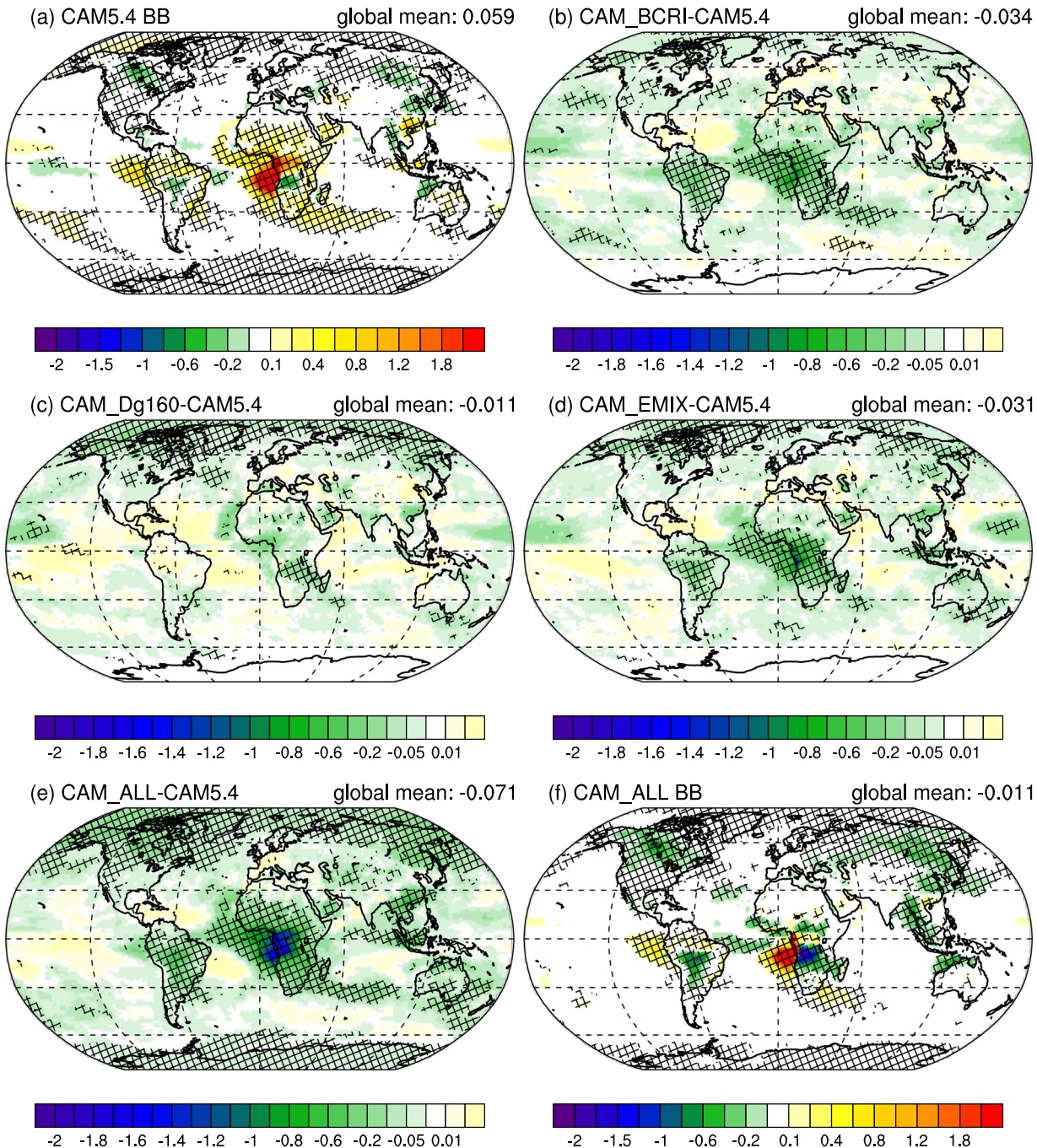

**Fig. 5 Modeled radiative effect due to biomass burning aerosol-radiation interactions. a** default CAM5.4 biomass burning (BB) radiative effect due to aerosol-radiation interactions (RE_ari), **b** the difference in RE_ari due to changes in BB black carbon refractive index (CAM_ BCRI – CAM5.4), **c** the difference in RE_ari due to increasing BB aerosol size (CAM_Dg160 – CAM5.4), **d** the difference in RE_ari due to treating fresh BB aerosol as externally mixed (CAM_EMIX – CAM5.4), **e** the difference in RE_ari due to all of the previous changes (CAM_ALL – CAM5.4), and **f** the RE_ari of BB with all of the previous changes (CAM_ALL). Hatching indicates regions where the change over the ensemble years is significant to the 0.05 level. Note difference in color bars.

forcing due to anthropogenic influence on BB emissions. In this work comparisons are made using the radiative effects of BB aerosols over the period 2003–2011 in the different CAM5.4 BB sensitivity tests. The focus is on the radiative effect due to aerosol-radiation interactions (RE_ari), defined in Ghan[63] as

$$RE_{ari} = \Delta(F - F_{clean}) \qquad (1)$$

where $F$ represents the TOA radiative flux, $F_{clean}$ represents TOA flux without aerosols, and $\Delta$ indicates the difference between two model simulations: with and without BB emissions.

Figure 5a shows BB RE_ari in the default model and panels b–e show the difference in BB RE_ari between different BB modifications and the default BB treatment. Figure 5f shows the BB RE_ari for CAM_ALL, which showed the most improvement in Fig. 4. The result of applying these microphysical changes to the model BB parameterization is a global reduction in RE_ari of 0.071 W m$^{-2}$, which causes the direct radiative effect of BB aerosol in CAM5.4 to transition from positive (0.059 ± 0.009 W m$^{-2}$) to slightly negative (−0.011 ± 0.011) (Table 2). Regionally these results are more pronounced, with a change in

**Table 2 The radiative effect due to aerosol-radiation interactions (RE$_{ari}$) of biomas burning (BB) aerosol in the CAM5.4 sensitivity tests. The CAM5.4 sensitivity tests are for different BB aerosol treatments: default CAM5.4 (CAM5.4), CAM5.4 with decreased BB black carbon refractive index (CAM_BCRI), CAM5.4 with increased BB aerosol size (CAM_Dg160), CAM5.4 with externally mixed, fresh BB aerosol (CAM_EMIX), and CAM5.4 with all of the previous changes (CAM_ALL). These RE$_{ari}$ are considered over the entire globe, the tropics, and the Arctic. Standard deviations from the 5 ensemble means are included.**

| Model Simulation | BB RE$_{ari}$ (W m$^{-2}$) Global | Tropics (25˚S-25˚N) | Arctic (60˚N-90˚N) |
|---|---|---|---|
| CAM5.4 | 0.059 (± 0.009) | 0.115 (± 0.020) | 0.023 (± 0.005) |
| CAM_BCRI | 0.025 (± 0.014) | 0.06 (± 0.024) | −0.001 (± 0.01) |
| CAM_DG160 | 0.048 (± 0.003) | 0.11 (± 0.011) | −0.03 (± 0.005) |
| CAM_EMIX | 0.028 (± 0.009) | 0.067 (± 0.014) | −0.016 (± 0.009) |
| CAM_ALL | −0.011 (± 0.011) | 0.013 (± 0.018) | −0.065 (± 0.006) |

RE$_{ari}$ of ~−2 W m$^{-2}$ in southern Africa BB regions and ~−0.5 W m$^{-2}$ in South American and Temperate BB regions (Fig. 5f). Arctic (60˚N–90˚N) and Tropical (25˚S–25˚N) BB regimes have changes in RE$_{ari}$ of −0.088 W m$^{-2}$ and −0.102 W m$^{-2}$, respectively (Table 2). Changes in the Arctic are driven largely by changes in size (CAM_DG160). This may be due to increased scattering effects in the Arctic, though there is little statistically significant change when looking at aerosol scattering optical depth (ASOD; Supplementary Fig. 18). A more likely scenario for these changes in the Arctic is enhanced dry and wet deposition due to increased aerosol size. This agrees with a decrease in Mode 4 POA and BC lifetimes in the Arctic from CAM5.4 (3.06 days for POA, 3.09 days for BC) to CAM_DG160 (2.37 days for POA, 2.39 days for BC) (Supplementary Table 12). Changes in the tropics are driven by decreases in BC RI (CAM_BCRI) and BC absorption enhancement (CAM_EMIX) especially over southern African cloud decks where cloud-top scattering can enhance absorption in above-cloud BB plumes.

One might assume that changes in SSA would be inversely correlated with RE$_{ari}$, and models with higher BB SSA would have lower RE$_{ari}$. However, this is not necessarily the case. CAM_BCRI, which has less than half the increase in global BB SSA (+ 2.1%) compared to CAM_EMIX (+ 4.6%) (Supplementary Table 10), has a similar RE$_{ari}$ to CAM_EMIX (Table 2); meanwhile, CAM_DG160, which has the same increase in global BB SSA (+ 2.1%) as CAM_BCRI (Supplementary Table 10), has about half the reduction in RE$_{ari}$ as CAM_BCRI (Table 2). This seeming incongruity can be explained by looking at BB aerosol absorption optical depth (AAOD) (Fig. 6).

The global average AAOD for CAM5.4 (1.07 × 10$^{-3}$; Fig. 6a) decreases by ~30% with all of the BB modifications in CAM_ALL (7.24 × 10$^{-4}$; Fig. 6f), corresponding with an overall increase in SSA. While CAM_BCRI and CAM_EMIX both show a reduction in AAOD, CAM_DG160 has very little change in AAOD, and, in the case of the southern Africa BB region, a significant increase in AAOD. This enhanced absorption is attributable to the larger, internally-mixed aerosol in this characteristically high BC emission region intercepting and absorbing more incoming solar radiation. While an increase in AAOD seems incongruent with the observed increase in SSA (Fig. 4), the fraction of AOD due to scattering also increases (Supplementary Fig. 18) resulting in an increase in BB SSA. From this comparison it can be concluded that, while scattering does increase the SSA of the smoke, attendant increases in absorption due to larger, high BC fraction aerosols counteract the reduction of the globally averaged RE$_{ari}$ due to scattering in CAM_DG160. Comparisons to AERONET observations show that CAM5.4 tends to underestimate AOD and AAOD in BB regions.

Increasing aerosol size (CAM_Dg160) improves the AOD and AAOD model performance in African BB regions (Supplementary Fig. 21, 22), though the tendency of GFEDv3 to underestimate BB emissions[25] may also explain this disagreement.

The similar RE$_{ari}$ global averages for CAM_BCRI and CAM_EMIX (Table 2) can be explained by the differences in lifetime of the modified BB aerosol. In CAM_BCRI, the RI of all BB BC from both Mode 1 and Mode 4 is changed. Conversely, in CAM_EMIX, the externally mixed calculation of optical properties only applies to Mode 4, the lifetime of which is about half that of Mode 1[43]. This reduces the distance over which the externally mixed BB aerosol with externally mixed optical properties can be transported, reducing its global coverage and subsequently its RE$_{ari}$. Given that Mode 4 has the highest contribution to increased BB SSA in the model, this helps explain the CAM_EMIX simulation's lower than expected effect on RE$_{ari}$. Extending the externally mixed optical property calculation to Mode 1 would further increase model SSA, likely improving agreement between model and observations in Fig. 4 and driving model BB RE$_{ari}$ further into the negative. This modification is not included in this study due to the more likely scenario that aged aerosols in Mode 1 will be internally mixed—leading to a BC absorption enhancement—and the lack of observational data for highly aged (multiple days) BB aerosol.

The agreement between CAM_ALL and observations is much better for CAM5.4 in SSA vs. BC:TC space, but still exhibits a lower SSA in higher BC:TC regions (Fig. 4). This may be partly explained by the previously mentioned lack of externally mixed BB aerosol optical properties in the accumulation mode. It could also be due to the greater number of available datasets from NAm having more of an influence on the observational mean.

## Discussion

Identification of underestimation in SSA of BB aerosol is an important step towards improving modeled aerosol optical properties and radiative forcing. While this analysis focused on CAM5.4, all of the models in this study showed some degree of SSA underestimation compared to observations. The biggest contributing factor to this disagreement is the internal mixing assumption in radiative calculations applied in every model except the externally-mixed GEOS-Chem simulations. Assuming that model aerosol is internally mixed is intended to represent the observed mixing states of these particles[19,32,64]. However, these idealized uniformly mixed (well-mixed RI) and perfectly coated (core–shell) aerosol treatments in models overestimate aerosol absorption enhancement[62]. Uniformly mixed treatments produce up to twice the absorption enhancement when compared to the varying mixing states of particles in a composition resolving aerosol model[33], and core–shell mixing can overestimate

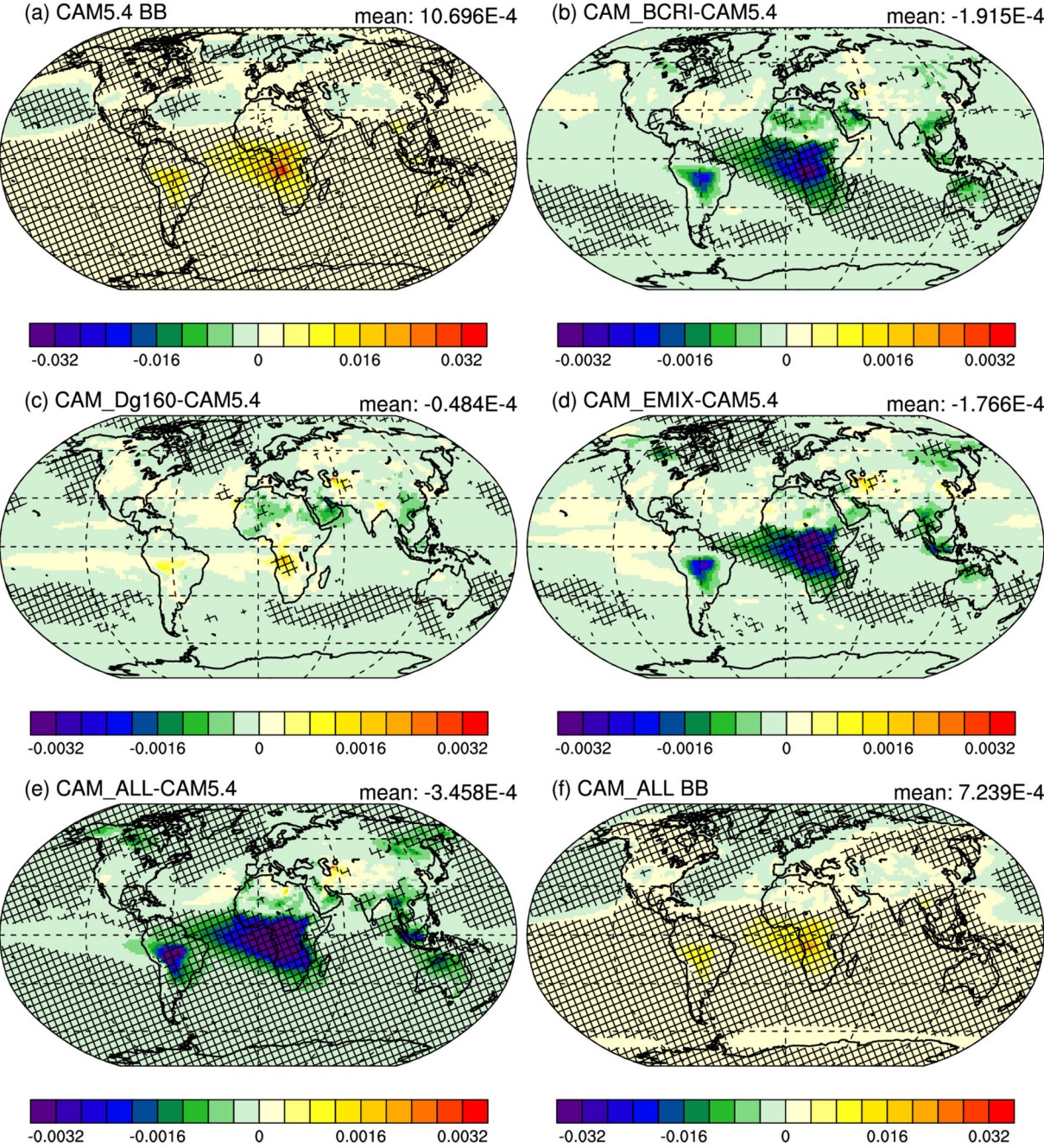

**Fig. 6 Modeled biomass burning absorption aerosol optical depth. a** default CAM5.4 biomass burning (BB) aerosol absorption optical depth (AAOD), **b** the difference in AAOD due to changes in BB black carbon refractive index (CAM_BCRI – CAM5.4), **c** the difference in AAOD due to increasing BB aerosol size (CAM_Dg160 – CAM5.4), **d** the difference in AAOD due to treating fresh BB aerosol as externally mixed (CAM_EMIX – CAM5.4), **e** the difference in AAOD due to all of the previous changes (CAM_ALL – CAM5.4), and **f** the AAOD of BB with all of the previous changes (CAM_ALL). Hatching indicates regions where the change over the ensemble years is significant to the 0.05 level. Note difference in color bars.

absorption by ~30% when compared to non-spherical aerosols with a non-uniform composition[19]. From the analysis of CAM5.4, we propose that an external mixture for BB aerosol optical calculations in ESMs and CTMs, while arguably not the most realistic approach to modeling mixing state, presents a more accurate approach to modeling optical properties of freshly-emitted BB aerosols. By modeling the optical properties of the more absorbing BC aerosol separate from the more scattering POA, the external mixing assumption of freshly emitted BB aerosol may help represent the sub-grid-scale variability in burn phase (smoldering versus flaming) observed in BB plumes[65,66], where smoldering is characterized by higher concentrations of scattering POA and flaming burns are characterized by higher concentrations of BC. This treatment can also be thought of as a temporary fix for necessary—but as of now, computationally limited—improvements to ESM and CTM mixing state for optical property calculations such as explicit particle shape and mixing treatments that deviate from spherical, core-shell, and well-mixed assumptions[34,67]; and particle resolved compositional diversity and heterogeneity in BC cores and particle coatings[33,34]. These findings indicate a need for

physical model treatments of aerosol composition that reduce absorption enhancement while representing hygroscopicity changes of aged, internally mixed aerosols.

Changes in hygroscopicity of particles with changes in mixing state can affect the cloud activation properties and lifetime of coated BC and, when coupled with changes in aerosol emission size, can add considerable uncertainty in calculations of BC $RE_{ari}$[31]. In this study, the external mixing for aerosol optical properties with the default internally mixed treatment of aerosol hygroscopicity minimizes these effects in CAM_EMIX, but the effects on cloud properties and aerosol lifetime in the model from changing aerosol size cannot be avoided in CAM_DG160 and CAM_ALL. These changes have the potential affect $RE_{ari}$ and the RE due to aerosol-cloud interactions by altering aerosol transport and deposition. Addressing uncertainty in these processes is also important, possibly leading to changes in BB $RE_{ari}$ of similar or greater magnitude than the optical property changes detailed in this study.

The effect of mass emission uncertainty of BB aerosols on CAM5.4 $RE_{ari}$ is not addressed in this paper, and is an important, but independent consideration. Studies utilizing CAM note model underestimations of BB aerosol optical depth[6] and BC mass concentrations at surface sites[61]. These studies, as well as this work, use the GFEDv3, though it has been shown that GFEDv3.1 underestimates fire emissions by ~11% when compared to GFEDv4[25]. Furthermore, Johnson et al.[38] found improvements in modeled AOD in HadGEM3 when GFEDv3 emissions were increased by a factor of 2. Recent work also shows a negative correlation between the change in AOD (at 550 nm) from preindustrial to present day and clear-sky $RF_{ari}$ across a wide range of models from AeroCom Phase II and the fifth Coupled Model Intercomparison Project (CMIP5)[3]. While different from this study in its focus on anthropogenic aerosols, the findings of Bellouin et al.[3] suggest that an increase in BB AOD associated with increased BB mass emissions would lead to a more negative BB $RE_{ari}$. In light of this study, an increase in BB emissions may lead to a more negative shift in global $RE_{ari}$ than the $-0.071\,\mathrm{W\,m^{-2}}$ observed in our CAM_ALL simulation. These are important considerations for further studies with CAM and other models in order to understand how changing mass emissions will affect microphysical changes and BB $RE_{ari}$.

This study contributes to an evolving understanding of carbonaceous aerosol climate effects, presenting evidence for a transition towards modeling BB aerosol with higher SSA. The current approach to modeling BB aerosols was driven by comparison to AERONET, satellite, and in situ observations, where underestimation in AeroCom model AAOD compared to AERONET data supported a movement toward increasing the emission of BB BC[1]. More recent work highlights a potential oversampling bias in AERONET when compared to in situ observations[68]. Additionally, the values of AAOD in Bond et al.[1] are larger than those in Kinne[69] due to stronger anthropogenic BC contributions and weaker mineral dust contributions in the former study. These more recent findings can help explain the lower model BB SSA when compared to observations in this study. Furthermore, many models focus on BB emissions from South American and African fires. While these fires have the largest emissions globally and warrant such attention in climate studies, caution should be taken in applying these aerosol optical properties on a global scale due to their characteristically lower SSA when compared to most temperate fuels and combustion conditions.

The models used in this study exhibit a wide range in BB $RE_{ari}$, from warming $(0.059 \pm 0.009\,\mathrm{W\,m^{-2}}$ (CAM5.4), $0.082\,\mathrm{W\,m^{-2}}$ (ECHAM6.3-HAM2.3, Supplementary Table 13), $0.05\,\mathrm{W\,m^{-2}}$ [16] (GEOS-Chem) to cooling $(-0.07$ to $-0.46\,\mathrm{W\,m^{-2}}$ [16] (GEOS-Chem)).

However, most of these models exhibit overestimated BB absorptivity in Fig. 2. Altering the CAM5.4 BB aerosol microphysical properties to better agree with observations resulted in a change of BB $RE_{ari}$ from positive to negative, with the most improvement in model SSA resulting from changes to BB aerosol mixing state. The expectation is that improving overall model agreement with observations in Fig. 2 through similar changes will reduce simulated BB $RE_{ari}$ across these models. This suggests that current estimates for BB $RE_{ari}$ are biased high. Improvements to other models may lower overall model $RE_{ari}$ in larger model ensemble datasets such as the sixth phase of the coupled model intercomparison project (CMIP6)[70] or the IPCC AR5[71]. The changes to $RE_{ari}$ of models will vary based on differences in microphysical treatments, and more work needs to be done to understand the responses in models to BB aerosol modifications. However, improvements to BB aerosols in models based on this SSA versus BC:TC framework have the potential to reduce bias in BB aerosol radiative forcing and establish BB aerosol as a cooling component in Earth's atmosphere.

## Methods

**Isolating biomass burning regions in the models**. Data for regions of strong biomass burning (BB) influence from each model was processed for direct comparison to observations. We isolated grid cells—varying in time, latitude, longitude, and height—where the mass concentrations of black carbon (BC) and organic aerosol (OA or OM) from BB ($BC_{BB}$ and $OA_{BB}$) made up greater than 85% of the total aerosol mass ($Tot\_Aer = \mathrm{Sulfate + Sea\ Salt + Dust + Secondary\ Organic}$ Aerosol (SOA) + BC + POA):

$$\frac{OA_{BB} + BC_{BB}}{Tot\_Aer} \geq 0.85, \qquad (2)$$

For CAM5, SOA is enhanced when $POA_{BB}$ and $BC_{BB}$ are present due to more semivolatile compounds partitioning into particle-phase on the surfaces of these primary aerosol. We take this into account in CAM5 by taking the difference between SOA mass concentrations from a model run with and a model run without BB aerosol and include this SOA contribution to $OA_{BB}$. This method results in more data points at low BC:TC ratios in the CAM5.3 simulation, likely due to its single size mode for fresh and aged BC and POA – which also includes SOA. For aerosols that have been transported long distances, SOA will contribute more to the aerosol mass as more SOA condenses on the primary aerosol and as $BC_{BB}$ and $POA_{BB}$ are depleted due to dry and wet deposition. This leads to an increased impact in the calculated total BB aerosol mass by SOA in these grid cells. Time resolution also plays a role, as the points that are present with daily temporal resolution are averaged out in the monthly CAM5.3 data (Supplementary Fig. 23).

Biomass burning regions from these calculations represent ambient conditions in the atmosphere, and cannot be considered dry for comparison with the in situ observations (relative humidities (RH) < 40%). Model simulations are reported as ambient due to limited RH data from the models. Inclusion of BB grid cells with higher RH will increase model SSA relative to model BB grid cells that are dry. As this is the case, we approach this comparison as an upper limit to the model SSA.

Different biomass regions were isolated based on 5 main biomass regimes: Temperate (30˚N–75˚N, 60˚E–300˚E), North America (18˚N–70˚N, 188˚E–306˚E), Africa (−30˚N–15˚N, 0˚E–60˚E), South America (−30˚N–0˚N, 280˚E–320˚E), and South East Asia (−10˚N–30˚N, 90˚E–130˚E). Supplementary Fig. 1 shows the isolated BB model surface concentrations for each model as well as the BB regions described in Fig. 2. Observational datasets are also plotted in Supplementary Fig. 1h and are described in detail in the supplementary.

**Isolating CAM5.4 mode aerosol properties from model biomass burning regions**. This study also makes use of aerosol microphysical properties from these regions. Aerosol total number concentration ($N_t$; cm$^{-3}$), dry number mode diameter (μm), volume extinction coefficient ($\beta_{ext}$; m$^{-1}$), and the wet refractive index ($n_w$) for each CAM5.4 aerosol mode was extracted from the lowest three-level heights (< 400 m) in the North America, Africa, and South America BB regions (defined in the previous section). Supplementary Fig. 2 shows the model BB regions and the observations used in Supplementary Fig. 3. Values for each modal $n_w$ were calculated based on an aerosol species volume-weighted mixture:

$$n_w = \frac{1}{V} \sum_{j=0}^{j} \frac{n_j q_j}{\rho_j}, \qquad (3)$$

Where j represents the species index, $n_j$ is the species refractive index, $q_j$ is the species mass mixing ratio, and $\rho_j$ is the species material density. V is the volume

mixing ratio, and is defined as

$$V = \sum_{j=0}^{j} \frac{q_j}{\rho_j}, \quad (4)$$

This treatment is described in more detail in Ghan and Zaveri[72] and is used in CAM to determine modal wet refractive indices[43].

This study focuses on the Mode 4 aerosol due to its dominance in BB regions. Once extracted, the mean, geometric, dry Mode 4 number diameter ($D_{gn,4}$), the Mode 4 total number ($N_4$), and the Mode 4 geometric standard deviation ($\sigma_{g,4} = 1.6$)[61] are used to plot the aerosol size distribution via the function

$$\frac{dN}{d \log D_p} = \frac{N_4}{(2\pi)^{1/2} \log \sigma_{g,4}} \exp\left(-\frac{\left(\log D_p - \log D_{gn,4}\right)^2}{2 \log \sigma_{g,4}}\right), \quad (5)$$

where $D_p$ is the midpoint diameter of each bin.

**Observational datasets and conditions.** This study drew from a variety of observational datasets of BB events. These datasets included aircraft field campaigns, ground-based observations, cookstove measurements, and burn laboratory measurements. The necessary data for this study were organic and inorganic aerosol concentrations, BC concentration, and extinction/absorption coefficients (for calculating SSA) near 550 and 700 nm wavelengths. When possible, Angstrom exponents were used to adjust scattering and absorption coefficients to 550 and 700 nm. When available, $N_t$ were used from observations. Organic and inorganic aerosol (OA + IA) concentrations were measured with either an aerosol mass spectrometer (AMS)[73], an aerosol chemical speciation monitor (ACSM)[74], or an offline EC/OC analyzer (Sunset Laboratories, Forest Grove, OR, USA); BC concentrations were measured with either a single particle soot photometer (SP2)[75], a multi-angle absorption photometer (MAAP)[76], or an offline EC/OC analyzer (Sunset Laboratories, Forest Grove, OR, USA); absorption coefficients were measured with either the Particle Soot Absorption Photometer (PSAP)[77], the MAAP, a three-wavelength photoacoustic soot spectrometer (PASS-3)[78], or the three-wavelength, University of Wyoming photo-acoustic spectrometer (UWPAS); scattering coefficients were measured using a nephelometer (NEPH)[79,80] or the PASS-3; extinction coefficients were measured with the cavity-attenuated phase shift particulate matter single scattering albedo (CAPS PMSSA) instrument; number concentrations were measured with either an ultra-high sensitivity aerosol spectrometer (UHSAS)[81], or a scanning mobility particle sizer (SMPS)/differential mobility particle sizer (DMPS)[82,83]. Other methods for collection of carbonaceous aerosol concentrations or emission factors used filter and thermal optical transmittance tests. Details for each dataset can be found in Supplementary Table 4.

Each observational dataset was chosen to correspond to BB influenced measurements (i.e., passes through plumes, BB in the near vicinity, or laboratory measurements of smoke). For the data processed in this study, further constraints were placed on the BB data. For aircraft measurements (i.e., SEAC⁴RS, DC3, ARCTAS, ORACLES), a lower threshold for BC concentrations of 0.9 μg m⁻³ was applied to designate high concentration passes through smoke plumes. For ground-based observations (i.e., Welgegund, GoAmazon), this threshold was lowered to 0.3 μg m⁻³ to account for the more dilute nature of the smoke in these campaigns compared to aircraft passes directly through the plume. The Welgegund data set was further constrained to time periods that were identified as being influenced by BB plumes. These times were identified in Vakkari et al.[66]. GoAmazon data was constrained to the AMS mass charge ratio 60 (f60) values of 0.01 or greater, which are representative of BB influenced samples[56]. All of the aircraft observations were also constrained to particular time periods that could be identified (e.g., in flight logs or in aircraft forward-facing cameras) as containing BB passes.

Aerosol measurements were also filtered to neglect aged smoke or the presence of other aerosol sources. This was done using AMS/ACSM inorganics data in a comparison similar to Eq. (2),

$$\frac{BC_{SP2} + OA_{AMS/ACSM}}{BC_{SP2} + OA_{AMS/ACSM} + Tot\_Inorg_{AMS/ACSM}} \geq 0.85, \quad (6)$$

where subscripts indicate the source instrument and $Tot\_Inorg$ = ammonium ($NH_4^+$) + sulfate ($SO_4^{2-}$) + chloride ($Cl^-$) + nitrate ($NO_3^-$) mass concentrations measured by the AMS/ACSM.

One final consideration for processing the observational data was the OM:OC ratio used to convert OM to OC for the scatter plot comparison. In the case of SEAC⁴RS and DC3 this value was reported in the AMS output. For ORACLES, the OM:OC ratio was calculated from the oxygen to carbon ratio (O:C) using the equation OM:OC = 1.260*O:C + 1.18[84]. For Welgegund, GoAmazon, and ARCTAS, where OM:OC data were unavailable, an OM:OC of 2 was assumed based on its similarity to the mean value OM:OC in other observational datasets.

The instruments used in this study have their own characteristic limitations that can lead to bias in SSA and BC:TC. Filter based measurements of absorption (PSAP, MAAP) and BC (MAAP) can be overestimated—especially in regions with high OA loadings—due to erroneous classification of multiple scattering as absorption[77,85,86]. Other non-linear responses can arise due to aerosol size distribution and increased filter loading.[87] NEPH data is sensitive to aerosol size

leading to a reduction in measured scattered light with a transition to forward and backward peaked scatter at sizes larger than ~300 nm[88,89]. AMS and ACSM data can underreport OA concentrations due to issues with aerosol collection and particle ionization efficiency[73,90]. SP2 measurements tend to underestimate BC mass concentrations if not corrected for the high concentrations encountered in BB smoke plume measurements[91]. This is in contrast to EC/OC analyzers, which can overestimate EC and underestimate OC due to misclassification of absorbing OC as EC[18]. PASS-3 and UWPAS exhibit small error in absorption coefficient arising from instrument dependent uncertainties (i.e., resonant frequency, resonator quality factor, microphone pressure, and laser power)[92]. Lastly, SSA calculated from UWPAS utilizes extinction coefficients calculated from the CAPS PMSSA[93], which suffers from a similar scattering bias at larger aerosol sizes as the NEPH.

Uncertainty estimates based on the various instrument adjustments, calibrations, corrections, and assumptions for all observations are added to the scatter plot data. Uncertainties from Pokhrel_2016 are described in more detail in Pokhrel et al.[94] and uncertainties from Liu_2014 are described in more detail in Liu et al.[95]. Uncertainties from the processed observational campaigns came from the data file notes or references therein. In the case of ORACLES and GoAmazon SSA, uncertainty was not reported in the data file notes. Here we assume uncertainties of 10% for the NEPH[96] and 20% for the PSAP[77] for these campaigns, calculating the SSA uncertainty via Gaussian error propagation. Yokelson_2009 data did not report SSA or BC:TC uncertainty. Here we assume a 5% uncertainty for SSA and a 40% uncertainty for BC:TC based on the average uncertainties from the other observational datasets. Lastly, the UWML_2017 uncertainties were calculated via Gaussian error propagation for BC (6% for absorption[93] and 16% for MAC_BC (16%)) and OC (20% for SMPS total aerosol volume and 38% for AMS aerosol density), calculating BC:TC uncertainty from these values in a similar fashion. SSA uncertainty was assumed to be 6% based on Foster et al.[93]. More details can be found in Supplementary Table 5.

**Lognormal fit to observed aerosol size distributions.** The observations were fit to a lognormal distribution after first deriving the geometric number mode diameter ($D_{gn}$) and the geometric standard deviation ($\sigma_g$) of the observed aerosol size distributions. $D_g$ was used as the midpoint diameter of the bin with the maximum aerosol counts. $\sigma_g$ was calculated following the TSI Incorporated Application Note PR-001 (https://www.tsi.com/getmedia/1621329b-f410-4dce-992b-e21e1584481a/PR-001-RevA_Aerosol-Statistics-AppNote?ext = .pdf),

$$\sigma_g = 10^{\left[\frac{\sum n_i \left(\log D_i - \log D_{gn}\right)^2}{N-1}\right]^{1/2}}, \quad (7)$$

where i represents the bin index, $n_i$ is the number in bin i, $D_i$ is the midpoint diameter in bin i, and N is the total number of particles summed over all bins. Once we calculate $\sigma_g$, we calculate the size distribution similar to Eq. (5). We also apply this fitting to minimum and maximum counts for each bin, with the range represented by the color-fill about the size distribution fitting lines in Supplementary Fig. 3.

When applying this fit to the observations, all of the datasets were treated as unimodal within the size ranges of the instruments used. One exception to this was the Welgegund data from Vakkari et al.[97], which was tri-modal within the size range.

Observed and modeled size distribution comparisons are presented in Supplementary Fig. 3.

**Model simulations.** Detailed information regarding the model simulations can be found in Supplementary Tables 1 and 2. Each CAM5.4 simulation consists of 5 ensembles that are varied by applying different initial temperature perturbations of the order 10⁻¹⁴ [98]. Each ensemble is initialized by a spin-up year that is neglected in the analysis. Further modifications are made to BC RI (CAM_BCRI), BB aerosol size (CAM_DG160), and BB aerosol mixing state (CAM_EMIX). These modifications are made to the freshly emitted BB aerosol (Mode 4), which represents emitted POA and BC aerosol from combustion (e.g., biomass, biofuel, and fossil fuel sources) and can deposit out of the atmosphere or age into the accumulation mode (Mode 1) due to coating of sulfate and SOA[43,61]. Mode 4 made up a majority of BB aerosol in BB grid cells (>90% of BB aerosol number, >54% of BB aerosol extinction (Supplementary Table 3)). With global lifetimes of 2.78 days for Mode 4 BC and 2.77 days for Mode 4 POA (Supplementary Table 12), Mode 4 also serves as a reasonable comparison to observational datasets, which measure smoke with ages on the order of hours to a few days (Supplementary Table 4).

BC RI is treated as a constant over the defined shortwave bands (roughly 0.232–8 μm) in CAM5.4. Our modification is made to the offline physical property files for BC that are then read into the model.

In the CAM_DG160 simulation we modified the number emissions ($E_{number}$) of Mode 4 BB aerosols, manually reducing the size of the Mode 4 aerosol. In CAM, the emitted volume-mean aerosol diameter ($D_{emit}$) is determined from number emissions ($E_{number}$), aerosol density ($\rho$), and aerosol mass emissions ($E_{mass}$),[43]

$$D_{emit} = \left(\frac{E_{mass}}{\frac{\pi}{6}\rho E_{number}}\right)^{1/3}, \quad (8)$$

$D_{gn}$ is in turn calculated from $D_{emit}$ based on their lognormal distribution relationship using a fixed modal $\sigma_g$[43]

$$D_{gn} = \frac{D_{emit}}{\exp\left(1.5\left(ln\left(\sigma_g\right)\right)^2\right)}, \qquad (9)$$

The CAM5.4 global average BB $D_g$ (96.6 nm) is smaller than the observational global average BB $D_g$ (160.1 nm) (Supplementary Table 7). To change $D_g$ in the model to match observations, we decrease the model $E_{number}$ while leaving $E_{mass}$ unchanged. This partitions the available aerosol mass to a fewer particles, increasing their size. A scaling factor of $(100/160)^3$—or approximately 0.25—was applied to $E_{number}$ to simulate a global average $D_g$ from the model (159.1 nm) that is within 1% of the observational average $D_g$ (160.1 nm) (see Supplementary Tables 7 and 8).

The CAM_EMIX simulation is the same as CAM5.4 with the addition of externally mixed Mode 4 BB aerosols in the modal aerosol optical property treatment. The current treatment of aerosol optical properties treats each of the externally mixed modes (accumulation, Aitken, coarse, primary carbon) as an internal mixture of aerosol species in that mode, calculating the volume-weighted refractive index for that mode (see Eqs. 3 and 4). Using this refractive index and the modal aerosol radius the model then calculates the specific scattering and absorption coefficients, in addition to the asymmetry parameter for each mode, and combines the modal values to give the optical properties for all aerosols[61,72]. Our modification treats the primary carbon mode (Mode 4) optical property calculations as externally mixed BB BC and POA, but leaves anthropogenic BC and POA (i.e., biofuel and fossil fuel emissions) as an internal mixture. We calculate new aerosol diameters for the externally mixed BB BC and POA as an input to the optical property calculations, but leave the preexisting, 4-mode microphysical treatments unchanged in the model. The aerosol calculations follow Eqs. 8 and 9, with the same Mode 4 $E_{number}$ applied to all of the new species (POA BB, BC BB, and POA + BC from anthropogenic sources).

**Statistical Analysis**. For the observational data used to validate the model, we use a bivariate regression[99] when calculating a linear fit to the data. We also calculate relative standard error (RSE) to justify our use of a linear fit to the data. Uncertainty bounds for each dataset are included when available, and these values are described in Supplementary Table 5.

For Figs. 5 and 6, we use a two-tailed t-test for the 45 simulated model years (5 ensembles, with 9 years per ensemble) to determine points where the change is significant to the 0.05 level (hatching). Global-mean standard deviations in Table 2 are calculated from the 5 ensemble means.

## Data availability

AeroCom model data can be obtained from http://aerocom.met.no. Observational data sets can be accessed from the following sites: GoAmazon (https://www.arm.gov/research/campaigns/amf2014goamazon); ORACLES (https://espo.nasa.gov/oracles/archive/browse/oracles); SEAC⁴RS, DC3, and ARCTAS (https://www-air.larc.nasa.gov). Welgegund data can be requested from Ville Vakkari (Ville.Vakkari@fmi.fi) or Paul Beukes (paul.beukes@nwu.ac.za).

## Code availability

Source codes and model setups needed to repeat all CAM5.4 model simulations along with output files from the model runs are available on request from our corresponding author (X.L.).

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

## Acknowledgements

This work was supported in part by: the U.S. Department of Energy (DOE), Office of Science, Office of Workforce Development for Teachers and Scientists, Office of Science Graduate Student Research (SCGSR) program, the "Enabling Aerosol-cloud interactions at GLobal convection-permitting scalES (EAGLES)" project (74358) funded by the DOE Office of Science, Office of Biological and Environmental Research, Earth System Model Development (ESMD) program, and the DOE as the NSF, DOE, and USDA Joint Earth System Modeling (EaSM) Program. The authors would like to acknowledge the use of computational resources (doi:10.5065/D6RX99HX) at the NCAR-Wyoming Supercomputing Center provided by the National Science Foundation and the State of Wyoming, and supported by NCAR's Computational and Information Systems Laboratory. Data from the GoAmazon field campaign were obtained from the Atmospheric Radiation Measurement (ARM) user facility, a U.S. Department of Energy (DOE) Office of Science user facility managed by the Biological and Environmental Research program (https://www.arm.gov/research/campaigns/amf2014goamazon). Data from the ORACLES field campaign were obtained from the NASA Ames Earth Science Project Office (ESPO) data archive (https://espo.nasa.gov/oracles/archive/browse/oracles). Data from the SEAC4RS, DC3, and ARCTAS field campaigns were obtained from the NASA Langley Research Center (LaRC) Airborne Science Data for Atmospheric Composition data archive (https://www.air.larc.nasa.gov). Art Sedlacek added valuable input interpreting observational data and giving direction in finding documentation for the GoAmazon field campaign. Bob Yokelson contributed data from the MILAGRO field project. Andrew Grieshop contributed observational data from cookstove measurements in India. Steven Howell provided guidance regarding the use of ORACLES data. Scot Martin and Suzanne de Sá provided guidance and AMS data for the GoAmazon field campaign. Funding from the University of Helsinki and the Finnish Meteorological Institute, as well as the Biogeochemistry Research Infrastructure Platform (BIOGRIP) of the Department of Science and Innovation of South Africa made measurements at Welgegund possible. D.W.-P. and P.S. acknowledge support from the NERC NE/L013746/1 (CLARIFY-2016) and NE/J024252/1 (GASSP) projects. The ECHAM-HAM simulations were performed using the ARCHER UK National Supercomputing Service. The ECHAM-HAMMOZ model is developed by a consortium composed of ETH Zurich, Max Planck Institut für Meteorologie, Forschungszentrum Jülich, University of Oxford, the Finnish Meteorological Institute, and the Leibniz Institute for Tropospheric Research, and managed by the Center for Climate Systems Modeling (C2SM) at ETH Zurich. P.S. acknowledges funding from the NERC CLARIFY NE/L013479/1 and A-CURE NE/P013406/1 projects AND THE European Research Council project RECAP with grant agreement 724602. R.B.S and G.M. acknowledge support from the Norwegian Research Council project SUPER (no. 250573). S.L.'s research performed as a post-doctoral fellow with Allison Aiken and Manvendra Dubey at Los Alamos National Laboratory funded by Department of Energy Office of Biological and Environmental Research Atmospheric System Research program is acknowledged.

## Author contributions

H.B., X.L., R.P., S.M., and Z.L. conceived and designed the research. H.B. ran the Community Atmosphere Model simulations, processed and analyzed observational and model data, and wrote the paper. T.M., H.K., T.B., G.M., R.B.S., D.W.-P., P.S., B.J., N.B., M.S., and R.S. contributed model simulation data and data interpretation. R.P., S.M., V.V., J.P.B., P.G.Z., S.L., and D.C. contributed observational data and data interpretation.

## Competing interests

The authors declare no competing interests.
