## [Peer Review File · Nature Communications]

REVIEWER COMMENTS

Reviewer #1 (Remarks to the Author):

This study presents some interesting comparisons between models and observed optical properties of carbonaceous aerosol from fires and the interplay between assumed properties, AAOD, and DRE. However, there is little observational support for the modeling hypotheses put forth to support the changes made to the model to engineer better agreement with the observations. In addition, the overall impact on the radiative effects are modest, given the other factors contributing to the uncertainties in fire radiative effects (a change of 0.07 W/m² in this study, given a range of biomass burning of 0.40 W/m² in the last IPCC). Finally, some of the results of the study are not sufficiently explored (described below). Thus in terms of clarity of results, potential impact, and the short format limitations, this paper is not suitable for publication in Nature Communications.

Major comments:

1. The differences between the models and their varying performance compared to observations of SSA shown in Figure 2 are not sufficiently discussed. The intermodal spread (as well as the spread within ECHAM-3-HAM2.3 and perhaps CAM5.3) appear comparable to the spread in the one model (GEOS-Chem) where the treatment of aerosol absorption and mixing state differs. It seems that these differences should be well understood prior to putting forth a hypothesis for improving the CAM5.4 performance.
2. The role of brown carbon (BrC) is not adequately addressed. A discussion of how important BrC absorption is at the 550 nm wavelength explored in Figures 2 and 3 is needed. For example, the difference between the GEOS-Chem simulations with and without BrC seem comparable (or larger) than many of the parameter modifications presented in Figure 4. It therefore seems incorrect to suggest that BrC is not among the main causes for the difference between model and observations (Line 220). The recent work on how models treat BrC (including photochemical whitening), and how this compares to the simulations shown in this study should also be discussed. Finally, given that the conclusions are largely based on SSA observations at wavelengths ≥ 550 nm, the role of absorption at shorter wavelengths (including the more prominent role of BrC), is not constrained in this study, and thus the title may be overstated.
3. Neither the main text nor methods discuss the potential measurement biases that might impact these results. Filter-based measurements of absorption are known to be susceptible to biases associated with multiple scattering and filter saturation. CAPS instruments also have some limitations at higher SSA and particle sizes. Foster et al., AMT, 2019 (which shares two co-authors with this study!) provide a discussion of some of these issues. These issues may be a major factor in model-measurement disagreement and should be discussed.
4. Line 133: Given the interannual variability in fire locations (in addition to magnitude), the use of a single model year to compare to observations from different years, could be an important limitation of this work. Fires in a given year may be from different ecosystems with different emission factors, and thus different SSA. Model-measurement comparisons here are not apples-to-apples. Additional work is needed to justify whether such comparisons are robust to interannual variability in emissions (and meteorology).
5. Given the issues raised above and in minor comments below (mismatch in model/measurement year, potential measurement biases, dry vs wet sampling, role of BrC) it's not clear that the changes proposed in Figure 4 are the leading candidates responsible for the model underestimate of SSA. These sensitivity tests are interesting as a hypothesis, but engineering the model-obs match with these changes may be "getting the right answer for the wrong reason", calling into question the DRE calculations that follow.

Minor comments

1. The abstract should be more specific about what observations and models were used, and the aerosol properties that were modified in CAM5 to improve agreement with observations. The final sentence of the abstract should be removed as this was not demonstrated in this study.

2. Table 1: dates of campaigns should be a range (e.g. ARCTAS is indicated as a single day)
3. Lines 118: Based on the methods, the authors looked at the total of fine particle non-refractory mass reported by AMS/ACSM plus BC from SP-2. This neglects the contribution of dust and larger sizes, and is thus not total "aerosol mass" as stated. It's also not clear in this section whether the 85% criteria is for campaign averages or point-by-point.
4. Lines 138-139: It doesn't seem sensible to use two different cut-offs (which are also so similar) in the model and observations, leading to a different subset of points. Ideally the model would be matched to the observations to provide an apples-to-apples comparison (see above).
5. Figure 1: it would be useful to match the colors used here to the colors used in Figure 3, i.e. all North American campaigns in shades of green, so that the reader can compare the observed values with relevant model ranges in Figure 3 (i.e. on line 150 is the regional dependence of BC:TC and SSA similar in model and observations? Similarly, the statement on line 168 regarding the model:obs spread in BC:TC is currently not evident from the figures).
6. Line 145-146: the statement "modeled SSA is lower than observed in most cases" is incorrect. The slope is lower, but at low values of BC:TC, the model SSA is not consistently lower than observations.
7. Lines 148-150: the role of water in these comparisons needs to be quantified to understand how important its role is in model-measurement comparisons. Perhaps the authors could show both the dry and wet SSA in the CAM5.4 model to illustrate.
8. Figure 3: why doesn't this figure have the same set of models as in Figure 2?
9. Lines 207: "indicating an overestimation on BB aerosol absorption" seems to contradict the earlier statement on line 2020 and lines 209-210 which suggest that the difference in the SSA is due to an underestimation of scattering. Similarly, lines 213-214 seems to focus on the relative model overestimation of absorption, but the comparisons of Figure 3 seem to more specifically support the absolute underestimation of scattering.

Reviewer #2 (Remarks to the Author):

This paper argues that, as the title suggests, biomass burning aerosols in current climate models tend to lead to too much absorption of sunlight. The result is perhaps important to the climate modeling community, because the lingering uncertainty in model sensitivity to increasing greenhouse gas forcing is commingled with our uncertainty in the amount to which aerosols may be masking greenhouse warming by enhancing scattering of sunlight. It is the second of those uncertainties where this paper is arguing we have a systematic bias in models. However, the paper does not quite get to exactly the uncertainty in radiative forcing of biomass burning aerosols. The novel contribution of the paper is documenting a systematic bias in the intrinsic optical properties of biomass burning particles in models and helpfully points to some aspects of the parameterization of these intrinsic properties that might be adjusted to better match in-situ observations of these properties. The paper may be suitable for publication with some revisions as suggested below.

The main evidence for aerosols that are too absorbing is a comparison of scatter plots of single-scattering albedo of aerosols against the ratio of black carbon mass to total carbon mass in field campaign observations and several models for locations dominated by biomass burning aerosols. They also show the mass absorption coefficient relative to the black carbon to total carbon ratio. Both indicate that the optical properties of the biomass burning particles is more absorbing in the model compared to observations. Curiously, the paper does not discuss quantitatively at all the net biomass burning radiative forcing of the models compared against the observations in figures 2 and 3. They do note that the IPCC estimates net radiative forcing at $0.0 \pm 0.2 \text{ W m}^{-2}$. And they find that adjustments to aerosol parameterizations in one model lead to forcing ranging from 0.06 to -0.01 W m^{-2} , which is within the range of uncertainty of the IPCC estimate. I was unclear whether the implication of the paper is that the IPCC value is likely too positive and should be revised downward? Or whether the expected value of the IPCC estimate might be reasonable, but that the models under consideration in this paper have forcing values that are more positive? Or perhaps the models may get

a net forcing value that is reasonable, but do so for the wrong reasons, since the intrinsic aerosol properties are not correct? As correctly noted by the authors, there are a host of other factors that affect the net forcing that are not addressed here. The second half of the paper is dedicated to the sensitivity of the net radiative forcing in one model to the aerosol parameterizations, so the paper would be more impactful if the overall implications could be more specifically stated. In particular, this would help clarify whether this paper is merely showing that uncertainty in biomass burning aerosol properties is contributing to the wide uncertainty in biomass burning radiative forcing, or whether the models are truly overestimating aerosol absorption, as the title is boldly stating.

It is also very clear from looking at figures 2 and 3 that the models do a poor job of capturing the range and distribution of the ratio of black carbon mass to total carbon mass (BC:TC), which is attributed by the authors to the emissions, which are not adjusted in the experiments presented in the paper. While the sensitivity studies do indicate that CAM5 can be adjusted to yield a more realistic dependence of single-scattering albedo on the BC:TC ratio, and that doing so yields a more positive net radiative forcing of biomass burning aerosols. Presumably, one could achieve a similar result, at least qualitatively, on the net radiative forcing by adjusting the emission to reduce the BC fraction of the carbon emitted from biomass burning. Is there an indication of whether the magnitude of the result in net radiative forcing might be comparable from such an adjustment? If the conclusion of the paper is that the specific choices for the parameterization of the size, mixing state and refractive index of biomass burning aerosols impacts the total absorption, is it possible that the sensitivity of the models to these details would be weaker if the BC:TC ratio were more similar to the observations?

There is little discussion of the uncertainties in the observations represented in figure 1 (and included in figures 2-4). There are citations in the supplementary tables, however, it would be good to include a statement in the body of the paper explaining what these error bars represent. Are these just variability in the measurements, instrument uncertainties, or combinations of these uncertainties?

The changes in net radiative forcing in the altered versions of the CAM5 model discussed in the paragraph beginning line 351 will depend on the albedo of the surface and/or clouds beneath the biomass burning aerosol. Are there responses in the clouds due to the difference in the aerosol properties between these simulations that are likely to have a meaningful impact on the brightness of the surface beneath the biomass burning aerosol?

The statement in lines 447-448 that "assuming internal mixtures for aerosols in models has been considered a more realistic approach" needs a reference. That this particular factor seems to be impactful on the relationship of SSA to BC:TC seems to be an important result from this paper.

Reviewer #3 (Remarks to the Author):

The authors use simulations from the phase III of the aerocom model intercomparison exercise and compare them to a range of observational data to derive the conclusion that the biomass-burning aerosol optical properties in atmospheric composition models are generally too absorbing. They focus on CAM5, which they use to test a range of proposed changes in the BB aerosol refractive index and assumed size distribution in order to reduce this bias. The manuscript is very well written and presented, and the authors master their subject very well. An impressive range of observational datasets are used in a very synthetic way (by using scatterplots of SSA vs BC/TC ratio) to assess simulated optical properties; the authors have been careful to restrict this observational dataset to their species of interest. The fact that several well known models are evaluated add credence to their claim about the BB aerosol properties being too absorbing. Using SSA vs BC/TC ratio is a pertinent metric as it allows the authors to focus exclusively on the aerosol optical properties and not on the fact that BB emissions are over or underestimated. The proposed small scale changes in the BB aerosol optical properties are easy to implement and very interesting to the atmospheric composition

modelling community. I recommend the manuscript to be accepted and have only minor questions and remarks:

- How does the work on the lead author on Brown Carbon (Brown et al., 2018, ACP) fit in this study? This is mentioned in passing but I think it deserves a more developed discussion. In this work, brown carbon is apparently not used, is there a reason for this?

- This is linked with the previous question: the manuscript focuses on the absorption in the visible part of the spectrum (550nm). What about other wavelengths, and in particular the near UV where brown carbon could make a difference?

- This work is of course of interest to climate model (and the extension about radiative forcing is of special interest there), but also to general atmospheric composition modellers. This could be mentioned, either in the title or in the abstract.

- What is the impact of the proposed changes on extinction? This is mentioned I think in the discussion about the impact on radiative properties, but I think a Figure showing global AOD and AAOD as simulated by CAMS5 with and without the proposed changes in aerosol optical properties would be very interesting to the reader.

- While the BB emissions in themselves are indeed out of the scope of this paper, it would be nice to have possibly some direct comparison of simulated values against AERONET for example (for AAOD and AOD if relevant), to check if the proposed changes do have a positive impact against AERONET. Of course, this evaluation would be impacted by possible defaults (and underestimation) of GFEDv2, which the authors mentioned already.

Kind regards, and congratulations for the impressive work.
Samuel Remy

We thank the editor and the anonymous reviewers for the encouraging comments and constructive suggestions on the manuscript. Below, we explain how the comments and suggestions are addressed and make note of the revisions in the revised manuscript. The Reviewer and editor comments are in **blue**, our replies are in **black**, and the corresponding changes to the manuscript are in **red**. New references are cited by name in the text and included at the bottom of this response.

Reviewer #1

General Comments

This study presents some interesting comparisons between models and observed optical properties of carbonaceous aerosol from fires and the interplay between assumed properties, AOD, and DRE. However, there is little observational support for the modeling hypotheses put forth to support the changes made to the model to engineer better agreement with the observations. In addition, the overall impact on the radiative effects are modest, given the other factors contributing to the uncertainties in fire radiative effects (a change of 0.07 W/m^2 in this study, given a range of biomass burning of 0.40 W/m^2 in the last IPCC). Finally, some of the results of the study are not sufficiently explored (described below). Thus in terms of clarity of results, potential impact, and the short format limitations, this paper is not suitable for publication in Nature Communications.

Reply: We thank the anonymous reviewer for the detailed and thoughtful review of our work. We address their concerns in the following response and in the revised manuscript. Questions regarding observational support are discussed in Major Comment 5, and we clarify these points in the revised manuscript. Regarding the overall impact of the radiative effects, it is true that 0.07 W m^{-2} is small (although still represents $\sim 20\%$ of the uncertainty range), but the key finding is elsewhere: our study shifts the fire aerosol radiative forcing towards a negative sign, which will have important implications for the overall anthropogenic forcing assessed by the IPCC. With the increasingly intensive fires in some regions of the globe, our study of understanding the optical properties of carbonaceous aerosol and quantifying their radiative effects potentially has large impacts. We have also explored the results of our study in more detail (e.g., differences between the models, effects of aerosol water) in our response to Major Comment 1. Thus, we have significantly improved the clarity of results in the revised manuscript.

We address their comments below.

Major Comments:

1. The differences between the models and their varying performance compared to observations of SSA shown in Figure 2 are not sufficiently discussed. The intermodal spread (as well as the spread within ECHAM-3-HAM2.3 and perhaps CAM5.3) appear comparable to the spread in the one model (GEOS-Chem) where the treatment of aerosol absorption and mixing state differs. It seems that these differences should be well understood prior to putting forth a hypothesis for improving the CAM5.4 performance.

Reply: We have added additional discussion in the revised manuscript regarding the differences in SSA between the models based on analyses of comparing SSA and BC:TC versus model height and the effect of relative humidity (RH) on SSA. We also explore the effect of CAM5.3 temporal resolution and aerosol treatment on data points that appear at low BC:TC ratios (<0.04) shown in Fig. 2c.

For RH effects, we include a comparison of ambient and dry ($RH < 40\%$) biomass burning (BB) aerosol in the supplementary for CAM5.4. We use $RH < 40\%$ as our “dry” cut-off based on recommendations in the reference WMO/GAW (2016). We add the following figure to the supplementary material, which shows higher SSA grid cells and much of the Northern Asia (NAs) and North America (NAM) grid cells corresponding to $RH > 40\%$:

Supplementary Figure 12: Same as Fig. 2, but comparing the effect of grid cell relative humidity on SSA. Panel (a) is the same as Fig. 2a, while panel (b) shows an additional comparison where BB aerosols are processed to only include aerosol at relative humidity less than 40%.

For the height analysis of SSA and BC:TC, we compared the two ECHAM6.3 models, both of which show regions of low/high of SSA/BC:TC at upper model levels (lower pressure levels). We add the following figure to the supplementary material:

Supplementary Figure 14: Comparison of model pressure level vs. BC:TC and model pressure level vs. SSA at 550 nm for both ECHAM6.3-SALSA2 (top row) and ECHAM6.3-HAM2.3 (bottom row).

With Supplementary Figure 14 and the known aerosol model treatments in models (Supplementary Table 1), we add a few explanations for the SSA variation. First, both ECHAM6.3-HAM2.3 and ECHAM6.3-SALSA2 use hydrophobic emissions for BB aerosol (Kokkola et al., 2018; Zhang et al., 2012). These hydrophobic aerosol – if they do not become hydrophilic due to accumulation of soluble compounds or coagulation – will have greater mobility due to their smaller size and reduced wet scavenging, potentially leading to more efficient vertical transport. Model emission heights likely also play a role in this vertical spread in SSA, as emissions that escape the planetary boundary layer (PBL) are more likely to have longer lifetimes and long-range transport. In Fig. 2h, GEOS-Chem, which homogeneously mixes BB emissions within the PBL, has far less vertical variation in SSA than ECHAM6.3-HAM2.3, which emits 25% of BB aerosol above the PBL (Val Martin et al., 2010).

Based on the above analyses, we added the following text in the revised manuscript (Line 145): “Spread in model SSA at a given BC:TC is related to aerosol water content in the grid cells, with larger SSA, and much of the NAM and NAs grid cells, related to higher aerosol water content (Supplementary Figure 12). For ECHAM6.3-HAM2.3 and ECHAM6.3-SALSA2, some of the points with lower SSA/higher BC:TC correlate with upper level grid cells in the model (Supplementary Fig. 14). This is partly attributed to the emission of hydrophobic aerosols in these two models^{45,46}, which may be more prone to vertical transport due to their reduced wet scavenging. Modeled BB emission heights could also play a role in this vertical spread in SSA, as emissions that escape the planetary boundary layer (PBL) are more likely to have longer lifetimes and undergo long-range transport (Martin et al., 2010).”

For CAM5.3 as commented by the reviewer, we include a comparison of CAM5.3 with and without including BB SOA. The figure we include in the supplementary is:

Supplementary Figure 15: Testing the effect of biomass burning (BB) SOA in CAM5.3, showing CAM5.3 without (a) and with (b) SOA BB (panel (b) is identical to Fig. 2c). SOA BB is calculated by the following: SOA (simulation with BB aerosol) – SOA (simulation without BB aerosol). This describes the SOA that condenses on emitted BB aerosol.

We added the following text in the revised manuscript (Line 171): “This may be partially explained by lack of SOA formation in the models, as the low BC:TC (<0.05) Afr, SAM, and SEAs model data from CAM5.3 (Fig. 2c) are examples of grid cells influenced by SOA formation on BB aerosol (Supplementary Figure 15). Lower observed BC:TC in SAM observations compared to most modeled SAM grid cells may be due to higher observed SOA formation rates⁵⁶ than are simulated in the models.”

Line 449: “For CAM5, SOA is enhanced when POA_{BB} and BC_{BB} are present due to more semivolatile compounds partitioning into particle phase on the surfaces of these primary aerosol. We take this into account in CAM5 by taking the difference between SOA mass concentrations from a model run with and a model run without BB aerosol and include this SOA contribution to OA_{BB}. This method results in more data points at low BC:TC ratios in the CAM5.3 simulation, likely due to its single size mode for fresh and aged BC

and POA – which also includes SOA. For aerosols that have been transported long distances, SOA will contribute more to the aerosol mass as more SOA condenses on the primary aerosol and as BC_{BB} and POA_{BB} are depleted due to dry and wet deposition. This leads to an increased impact in the calculated total BB aerosol mass by SOA in these grid cells. Time resolution also plays a role, as the points that are present with daily temporal resolution are averaged out in the monthly CAM5.3 data (Supplementary Fig. 23).”

We also include Supplementary Figure 22, which shows the comparison of monthly versus daily temporal resolutions of model output for three participating models. We discuss this figure in Major Comment 5.

2. The role of brown carbon (BrC) is not adequately addressed. A discussion of how important BrC absorption is at the 550 nm wavelength explored in Figures 2 and 3 is needed. For example, the difference between the GEOS-Chem simulations with and without BrC seem comparable (or larger) than many of the parameter modifications presented in Figure 4. It therefore seems incorrect to suggest that BrC is not among the main causes for the difference between model and observations (Line 220). The recent work on how models treat BrC (including photochemical whitening), and how this compares to the simulations shown in this study should also be discussed. Finally, given that the conclusions are largely based on SSA observations at wavelengths ≥ 550 nm, the role of absorption at shorter wavelengths (including the more prominent role of BrC), is not constrained in this study, and thus the title may be overstated.

Reply: We thank the reviewer for their comments. To address them, we include both simulations from Brown et al. (2018). These simulations are CAM5.4 with brown carbon (CAM_BrC) and CAM5.4 with brown carbon and photochemical bleaching (CAM_BrCbl). These are included in panel b of Figures 2 and 3:

Figure 2: Comparison of the observation data from Fig. 1a (in gray) to a variety of global climate models. The best-fit to the model data is represented by the solid blue line. The model data is from (a) this study, (b) Brown et al.¹⁷, (c-g) AeroCom Phase-III simulations, and (h) Saleh et al. (2015). They are (a) CAM5.4, (b) CAM5.4 (w/ BrC), (c) CAM5.3, (d) ECHAM6.3-SALSA2.0, (e) ECHAM6.3-HAM2.3, (f) HadGEM3, (g) OsloCTM2, and (h) GEOS-Chem. Model data is representative of BB influenced regions which are color coded in the model output and are specified on the global plot in panel (i). Observational data from the specific regions are averaged, and the regional average and range of the BC:TC observational data is included at the bottom of the plot (color coded squares). The two CAM5.4 simulations with BrC (b) represent brown carbon with and without photochemical bleaching (BrC and BrCbl, respectively). The four GEOS-Chem simulations (f) represent the four model simulations from Saleh et al. (2015): non-absorbing organics and externally mixed aerosols (NA+EM); absorbing organics and externally mixed aerosols (A+EM); non-absorbing organics and internally mixed aerosols (NA+IM); and absorbing organics and internally mixed aerosols (A+IM). Given that the CAM5.4 simulations were run for a number of years, and the AeroCom and GEOS-Chem simulations are for a single year, interannual variation in model slope and intercept is represented by dashed blue lines in panels a and b (long dash = BrC, short dash = BrCbl in panel b).

Figure 3: BB mass absorption cross-section (MAC_{BB} ; abs. coeff. / ($[BC] + [OA]$), $m^2 g^{-1}$) versus BC:TC ratio. Available MAC_{BB} observations are plotted from Fig. 1. The models are the same as in Fig. 2. The solid gray line is the best fit to observations, with dashed lines representing the 95% confidence intervals of the fit. The solid blue line is a best fit to the model data. Thick black lines constrain a reasonable range in BB MAC^{18} .

From the previous two figures, the addition of brown carbon in CAM5.4 exacerbates the disagreement between observation and model, lowering model SSA (Fig. 2b) and raising model MAC outside of the reasonable range for BB aerosols (Fig. 3b). The inclusion of the photochemical bleaching has the effect of raising the SSA in the Tropical BB regions (Afr, SAM, SEAs), though it still underestimates SSA compared to observations. The decreasing SSA with including BrC is also consistent with the GEOS-Chem results.

Following the reviewer's comment, we have also included wavelength comparisons for CAM5.4 and CAM5.4_BrC in the supplementary material:

Supplementary Figure 16: Same comparison as Fig. 2a, but comparing SSA versus BC:TC at three different wavelengths: (a) 400 nm, (b) 550 nm, and (c) 700 nm. Observations for panels (b) and (c) are the same as for Fig. 1, while observations for panel (a) are from Pokhrel et al. (2016), which are part of observations in b) and c).

Supplementary Figure 17: Same as Supplementary Figure 16, but using CAM5.4_BrC.

Supplementary Figure 16 shows that CAM5.4 underestimates SSA at 550 and 700 nm, but agrees better with observations at 400 nm wavelength. This is surprising given that these wavelengths are more affected by brown carbon, which CAM5.4 does not include. When brown carbon is included in this comparison (Supplementary Figure 17), the model shows enhanced SSA at all wavelengths compared to Supplementary Figure 16.

Our statement that brown carbon is not the main cause for the disagreement between model and observations referred to CAM5.4, which shows even more disagreement when brown carbon is included. This indicates that brown carbon does contribute to the disagreement, but it doesn't explain the preexisting model treatments responsible for disagreement in CAM5.4.

We added the following discussion in the revised manuscript (Line 212): “What this collection of models shows is that all of the models overestimate BB aerosol absorption relative to extinction to some extent. Some models perform better than others in this comparison, especially with smaller BB aerosol imaginary refractive indices and external mixing assumptions. Of the models included in this comparison, two CAM5.4 simulations and two GEOS-Chem simulations include the effects of BrC on aerosol absorption. CAM5.4 simulations with BrC from Brown et al.¹⁷ exacerbate the excessive absorption in the model, with photochemical bleaching effects (CAM5.4_BrCbl) slightly improving model-observation slope agreement. At shorter wavelengths (400 nm) where the effects of BrC absorption are greater, CAM5.4 agrees better with observations (Supplementary Fig. 16) than CAM5.4_BrC (Supplementary Fig. 17), which continues to exhibit lower SSA than observations. This comparison emphasizes that CAM5.4 SSA is too low at both 400 and 550 nm wavelengths, and including BrC in the model further reduces model SSA. In order to help understand the underlying causes of the BB absorption enhancement in the models, the following section will address areas of improvement in CAM5.4, and how these improvements affect BB SSA.”

3. Neither the main text nor methods discuss the potential measurement biases that might impact these results. Filter-based measurements of absorption are known to be susceptible to biases associated with multiple scattering and filter saturation. CAPS instruments also have some limitations at higher SSA and particle sizes. Foster et al., AMT, 2019 (which shares two co-authors with this study!) provide a discussion of some of these issues. These issues may be a major factor in model-measurement disagreement and should be discussed.

Reply: We thank the reviewer for the comment. We have added discussion of known instrument biases in our methods section. Most of the instrument biases discussed would result in underestimation of the SSA (due to overestimation of absorption or underestimation of scattering coefficient) and an overestimation of BC:TC ratio (overestimation of EC or underestimation of OA). We mention that these responses are taken into account in the error bars reported in Fig. 1, 2, and 3 in the main text.

Line 120: “Instrument uncertainty estimates for these data are represented by error bars in Fig. 1 (see Methods).”

Line 551: “The instruments used in this study have their own characteristic limitations that can lead to bias in SSA and BC:TC. Filter based measurements of absorption (PSAP, MAAP) and BC (MAAP) can be overestimated – especially in regions with high OA loadings – due to erroneous classification of multiple scattering as absorption (Bond et al., 1999; Lack et al., 2008; Petzold et al., 2005). Other non-linear responses can arise due to aerosol size distribution and increased filter loading (Weingartner et al., 2003). NEPH data is sensitive to aerosol size leading to a reduction in measured scattered light with a transition to forward and backward peaked scatter at sizes larger than ~300 nm (Anderson et al., 1998; Onasch et al., 2015). AMS and ACSM data can underreport OA concentrations due to issues with aerosol collection and particle ionization efficiency

(Canagaratna et al., 2007; Tiitta et al., 2014). SP2 measurements tend to underestimate BC mass concentrations if not corrected for the high concentrations encountered in BB smoke plume measurements (May et al., 2014). This is in contrast to EC/OC analyzers, which can overestimate EC and underestimate OC due to misclassification of absorbing OC as EC (Saleh, 2020). PASS-3 and UWPAS exhibit small error in absorption coefficient arising from instrument dependent uncertainties (i.e., resonant frequency, resonator quality factor, microphone pressure, and laser power) (Lewis et al., 2008). Lastly, SSA calculated from UWPAS utilizes extinction coefficients calculated from the CAPS PMSSA (Foster et al., 2019), which suffers from a similar scattering bias at larger aerosol sizes as the NEPH.

Uncertainty estimates based on the various instrument adjustments, calibrations, corrections, and assumptions for all observations were added to the scatter plot data.”

4. Line 133: Given the interannual variability in fire locations (in addition to magnitude), the use of a single model year to compare to observations from different years, could be an important limitation of this work. Fires in a given year may be from different ecosystems with different emission factors, and thus different SSA. Model-measurement comparisons here are not apples-to-apples. Additional work is needed to justify whether such comparisons are robust to interannual variability in emissions (and meteorology).

Reply: We thank the reviewer for the great comment. We have included some discussion of CAM5.4 model interannual variability with respect to fire emissions and meteorology and included a figure that plots individual years for CAM5.4 in the supplementary. These simulations were conducted for a period of 2003 to 2011 with yearly varying emissions and meteorology.

Supplementary Figure 11: Same as Fig. 2a but for yearly BB SSA vs BC:TC from CAM5.4. The slope and y-intercept of the linear fit (blue line) are included in each panel.

We also include the annual variation in slope and intercept in Fig. 2 for CAM5.4, CAM5.4_BrC, and CAM5.4_BrCbl (see Fig. 2a, 2b in Major Comment 2). While the fit to the data can vary largely depending on fire variability (as the reviewer has noted in their comment) and fewer model data points for each year, the lower SSA compared to observations persists, and regional emission factors result in similar SSA. This indicates that matching the slopes from single simulation years to multi-year observations results in some uncertainties due to differences in emissions and meteorology between the years, but pursuing a means to correct the underestimation of SSA is still valid.

We added the following text in the revised manuscript (Line 142): “Interannual variation in slope is included for CAM5.4, CAM5.4_BrC, and CAM5.4_BrCbl (Fig. 2a, 2b), and depends on the global distribution of BB emissions (Supplementary Fig. 11).

Underestimation of SSA in these models compared to observations is consistent year-to-year.”

5. Given the issues raised above and in minor comments below (mismatch in model/measurement year, potential measurement biases, dry vs wet sampling, role of BrC) it’s not clear that the changes proposed in Figure 4 are the leading candidates responsible for the model underestimate of SSA. These sensitivity tests are interesting as a hypothesis, but engineering the model-obs match with these changes may be “getting the right answer for the wrong reason”, calling into question the DRE calculations that follow.

Reply: We have conducted a number of analyses (most of which are described in the other major comments) and the main findings are described below:

As previously discussed (Major Comment 4), the underestimation of SSA by models does not change when considering the yearly variations of emissions and meteorology. Additionally, we examine the impact of temporal resolution of model output (monthly versus daily) on the comparison of model-observation. Comparison of daily and monthly temporal resolutions was conducted for CAM5.3, ECHAM6.2-HAM2.2 (earlier version of model not used in paper comparison, but used here due to availability of data), and HadGEM3. From the following figure it can be shown qualitatively that differences in slope and y-intercept are small between monthly and daily temporal resolution:

Supplementary Figure 23: Same as Fig. 2, but showing a comparison between monthly (upper row) and daily (bottom row) temporal resolution for CAM5.3 (a,d),

ECHAM6.1-HAM2.2 (b,e), and HadGEM3 (c,f). Only three models are presented in this comparison as these are the only simulations with both monthly and daily output.

As previously discussed (Major Comment 3), instrument uncertainty estimates for the observation data are represented by error bars in Fig. 1 and discussion of known instrument biases added in our methods section. Most of the instrument biases discussed would result in underestimation of the SSA and an overestimation of BC:TC ratio. Thus, measurement biases are not the main issue in the model-observation disagreement.

As previously discussed (Major Comment 1), relative humidity effects will result in lower SSA from simulations, enhancing disagreement between observations and models in Fig. 2.

As previously discussed (Major Comment 2), brown carbon has the effect of further decreasing SSA, indicating that this is not the main issue in the model-observation disagreement.

Regarding our model sensitivity tests:

- Mie calculations in the models make use of aerosol size distributions, aerosol composition (refractive index (RI)), and wavelength of light to calculate the aerosol optical properties in the models. We reasoned that these are the inputs that would lead to a disagreement in model SSA. This led us to the sensitivity factors in the paper: BB aerosol size, BC RI (only absorbing component in BB aerosols that affects RI), and aerosol mixing state (affecting the mixed aerosol refractive index in CAM5.4). Lastly, the use of BC:TC minimizes effects of aerosol concentration on the actual slope and y-intercept in the SSA vs. BC:TC framework. In this paper we try to avoid changes in aerosol concentration, but the following figure shows a test where we vary absorbing aerosol concentrations using CAM6 (the latest version of CAM with the same aerosol treatment as CAM5.4 we used; the earlier CAM versions are no longer supported by NCAR) by scaling biomass burning BC (BCBB) emissions by 0.5 (here we do not include the generation of multiple ensembles and it is only for the years 2003-2005). This comparison shows that, qualitatively, the changes in SSA with lower BC emissions are small at lower BC:TC, with a slightly larger effect on SSA in regions with higher BC concentrations. This decrease in SSA below the slope for the default simulation is attributed to a decrease in aerosol size with a decrease in BC emissions.

- To justify the CAM5.4 sensitivity experiments we conducted model-obs comparison analyses and compared CAM5.4 to other model treatments to identify causes for disagreement with observations. (1) To defend the size changes in the model, we compared a number of size distributions from observations to regional and global BB treatments in CAM5.4 (Supplementary Tables 7 and 8; Supplementary Figure 3). This comparison supports the increase in CAM5.4 model aerosol geometric mean diameter, and encourages the continued use of the prescribed BB geometric standard deviation (1.6) due to its similarity to that of the analyzed global BB size distributions (1.5). (2) The BC refractive index that is used in our CAM_BCRI simulations is part of a recommended range in BC refractive indices inferred from observations (Bond and Bergstrom, 2006). Some models use the highest values (CAM5.4) or middle values (ECHAM6.3, GEOS-Chem) of BC RI from this comparison, and Bond and Bergstrom (2006) make it clear that these values may be overestimations for BC refractive index. Other models derive their BB or BC refractive indices from BB observational campaigns (OsloCTM2, HadGEM3), which have lower real and imaginary parts of the RI than the lowest recommended value from Bond and Bergstrom (2006). Given the lower values of campaign-based BC RI and the original use of Bond and Bergstrom (2006) in CAM5.4, we chose to use the lowest recommended value from Bond and Bergstrom (2006). (3) Lastly, the use of externally mixed BB aerosol in CAM5.4 is a simplified solution to the overestimated BC absorption enhancement that arises from the simplified internal mixtures in climate models. Cappa et al. (2012) show measured aerosol absorption enhancement that is much lower in observations than calculated with the idealized core-shell Mie calculations used in climate models. Also, volume mixing assumptions (such as those used in CAM5.4) have a greater absorption enhancement than non-uniform aerosol internal mixtures based on aerosol 3-D imaging (Adachi et al., 2010).

We added the following text in the revised manuscript (Line 253): “**Aerosol Mixing State (CAM_EMIX):** Mode 4 BB aerosols are treated as externally mixed POA and BC.

This is based on evidence that idealized internal mixing assumptions used in climate models have been shown to overestimate BC absorption enhancement when compared to observations⁶² and lead to greater BC absorption enhancement than non-uniform internal mixtures based on observed mixed aerosol¹⁹. Additionally, GEOS-Chem simulations in Fig. 2f that assume externally mixed aerosols agree better with observational SSA. Accumulation mode (Mode 1) aerosols remain well-mixed internally (default, Supplementary Table 2) along with the Mode 4 anthropogenic aerosol, only treating the freshly emitted BB aerosol as externally mixed. This treatment is for radiative calculation purposes only; dry and wet deposition – in addition to cloud interactions – remain the same as in CAM5.4.”

Line 361: “Assuming that model aerosol is internally mixed is intended to represent the observed mixing states of these particles (Adachi et al., 2010; Liu et al., 2015; Martins et al., 1998). However, these idealized uniformly mixed (well-mixed RI) and perfectly coated (core-shell) aerosol treatments in models overestimate aerosol absorption enhancement (Cappa et al., 2012).”

Minor Comments:

1. The abstract should be more specific about what observations and models were used, and the aerosol properties that were modified in CAM5 to improve agreement with observations. The final sentence of the abstract should be removed as this was not demonstrated in this study.

Reply: We removed the last sentence of our abstract and identified the observational datasets and model simulations used in the study. We do not include specific details regarding all of the observations and model datasets due to a lack of space in the abstract. This seemed necessary to preserve the main findings of this study. For convenience, we have included the modified version of the abstract below:

“Uncertainty in the representation of biomass burning (BB) aerosol composition and optical properties in climate models contributes to a range in modeled aerosol effects on incoming solar radiation. Depending on the model, the top-of-the-atmosphere BB aerosol effect can range from cooling to warming. By relating aerosol absorption relative to extinction and carbonaceous aerosol composition from 12 observational datasets to nine state-of-the-art Earth System Models/Chemical Transport Models, we identify varying degrees of overestimation in BB aerosol absorptivity by these models. Modifications to BB aerosol refractive index, size, and mixing state improve the Community Atmosphere Model version 5 (CAM5) agreement with observations, leading to a global change in BB direct radiative effect of -0.07 W m^{-2} , and regional changes of -2 W m^{-2} (Africa) and -0.5 W m^{-2} (South America/Temperate). Our findings suggest that current modeled BB contributes less to warming than previously thought, largely due to treatments of aerosol mixing state.”

2. Table 1: dates of campaigns should be a range (e.g. ARCTAS is indicated as a single day)

Reply: Since it is necessary for the observational data to consist of BB plumes, we have isolated specific days from the selected campaigns that best represent within plume passes/ conditions. To make it clearer we have added a comment to the table caption explaining that these are single dates and not the entire campaign range.

“Table 1: Observational datasets from Fig. 1, along with their locations and sample collection dates. Single dates indicate that data was used from a single day of the campaign. Additional information can be found in Supplementary Tables 4 and 5.”

3. Lines 118: Based on the methods, the authors looked at the total of fine particle non-refractory mass reported by AMS/ACSM plus BC from SP-2. This neglects the contribution of dust and larger sizes, and is thus not total “aerosol mass” as stated. It’s also not clear in this section whether the 85% criteria is for campaign averages or point-by-point.

Reply: We no longer include this statement in the revised paper, and the 85% criteria is applied on a point by point basis. To take this in account, we have modified the sentence to read:

Line 112: “This data comes from 12 field and laboratory campaigns conducted in a number of source regions (e.g., North America, South America, Africa, India, Indonesia; Fig. 1, Table 1), with the criterion that BB carbonaceous aerosols dominate each data point (Methods).”

4. Lines 138-139: It doesn’t seem sensible to use two different cut-offs (which are also so similar) in the model and observations, leading to a different subset of points. Ideally the model would be matched to the observations to provide an apples-to-apples comparison (see above).

Reply: Thanks for the comment. We reprocessed the model simulations to reflect the same BB mass cut-off as the observations (85%). This is now reflected in the figures and the Methods in the revised manuscript:

(Line 441): “We isolated grid cells – varying in time, latitude, longitude, and height – where the mass concentrations of black carbon (BC) and organic aerosol (OA or OM) from BB (BC_{BB} and OA_{BB}) made up greater than 85% of the total aerosol mass ($Tot_Aer = Sulfate + Sea\ Salt + Dust + Secondary\ Organic\ Aerosol\ (SOA) + BC + POA$):

Eq. 2
$$\frac{OA_{BB} + BC_{BB}}{Tot_Aer} \geq 0.85$$
 ,

5. Figure 1: it would be useful to match the colors used here to the colors used in Figure 3, i.e. all North American campaigns in shades of green, so that the reader can compare the observed values with relevant model ranges in Figure 3 (i.e. on line 150 is the

regional dependence of BC:TC and SSA similar in model and observations? Similarly, the statement on line 168 regarding the model:obs spread in BC:TC is currently not evident from the figures).

Reply: Our intent was to represent the range in observational data at the bottom of Fig. 2 panels for model-obs comparison to allow for the separate observational datasets to be shown in Fig. 1. From our perspective, this approach allows the figures to build on one another better than using regional color coding for Fig. 1, which gives the same information as the means and ranges at the bottoms of each panel in Fig. 2. Furthermore, some observational datasets that are either outside of the global regional specifications (Greishop et al., 2017) or consisting of laboratory burn experiments that lack regional coding (Liu et al., 2014; Pokhrel et al., 2016), so a number of the observational datapoints in the regional figure remain unspecified (see following figure).

To regionally code the data points from Liu et al. (2014) and Pokhrel et al. (2016) will add considerable work that we believe will add little additional information to the paper. Therefore, we propose keeping our original observational BC:TC means/ranges in Fig. 2.

We have added a sentence to the text to bring more attention to the observational ranges in Fig. 2.

Line 166: “Regionally dependent BC:TC is also visible in the observational data, with regional means denoted at the bottom of each panel of Fig. 2.”

6. Line 145-146: the statement “modeled SSA is lower than observed in most cases” is incorrect. The slope is lower, but at low values of BC:TC, the model SSA is not consistently lower than observations.

Reply: We have removed in the revised manuscript: “, but modeled SSA is lower than observed in most cases”.

7. Lines 148-150: the role of water in these comparisons needs to be quantified to understand how important its role is in model-measurement comparisons. Perhaps the authors could show both the dry and wet SSA in the CAM5.4 model to illustrate.

Reply: See our reply to Major Comment 1.

8. Figure 3: why doesn't this figure have the same set of models as in Figure 2?

Reply: We updated this figure to include all of the models from Figure 2.

9. Lines 207: "indicating an overestimation on BB aerosol absorption" seems to contradict the earlier statement on line 202 and lines 209-210 which suggest that the difference in the SSA is due to an underestimation of scattering. Similarly, lines 213-214 seems to focus on the relative model overestimation of absorption, but the comparisons of Figure 3 seem to more specifically support the absolute underestimation of scattering.

Reply: We want to explain that models likely have some combination of size and composition issues, but those that agree well in the MAC comparison are likely to have model-observational disagreements in Fig. 2 that are due to size related issues. In either case (overestimated absorption, underestimated scattering), SSA will be lower which is what we see in almost all of the models in Fig. 2.

We modified the paper to make this clearer:

(Line 205): "Some models in Fig. 3 show a deviation from observations at higher BC:TC (Fig. 3a-d), indicating an overestimation in BB aerosol absorption. Models that show less divergence from observations (Fig. 3e-h) indicate a disagreement that is due in part to size dependent scattering differences between the model and observations."

References:

Adachi, K., Chung, S. H. & Buseck, P. R. Shapes of soot aerosol particles and implications for their effects on climate. *Journal of Geophysical Research: Atmospheres* 115, (2010).

Anderson, T. L. & Ogren, J. A. Determining Aerosol Radiative Properties Using the TSI 3563 Integrating Nephelometer. *Aerosol Science and Technology* 29, 57–69 (1998).

Bond, T. C., Anderson, T. L. & Campbell, D. Calibration and Intercomparison of Filter-Based Measurements of Visible Light Absorption by Aerosols. *Aerosol Science and Technology* 30, 582–600 (1999).

Bond, T. C. & Bergstrom, R. W. Light Absorption by Carbonaceous Particles: An Investigative Review. *Aerosol Science and Technology* 40, 27–67 (2006).

- Brown, H. *et al.* Radiative effect and climate impacts of brown carbon with the Community Atmosphere Model (CAM5). *Atmospheric Chemistry and Physics* **18**, 17745–17768 (2018).
- Canagaratna, M. R. *et al.* Chemical and microphysical characterization of ambient aerosols with the aerodyne aerosol mass spectrometer. *Mass Spectrom. Rev.* **26**, 185–222 (2007).
- Cappa, C. D. *et al.* Radiative Absorption Enhancements Due to the Mixing State of Atmospheric Black Carbon. *Science* **337**, 1078–1081 (2012).
- Foster, K., Pokhrel, R., Burkhart, M. & Murphy, S. A novel approach to calibrating a photoacoustic absorption spectrometer using polydisperse absorbing aerosol. *Atmos. Meas. Tech.* **12**, 3351–3363 (2019).
- Grieshop, A. P., Jain, G., Sethuraman, K. & Marshall, J. D. Emission factors of health- and climate-relevant pollutants measured in home during a carbon-finance-approved cookstove intervention in rural India. *GeoHealth* **1**, 2017GH000066 (2017).
- Kokkola, H. *et al.* SALSA2.0: The sectional aerosol module of the aerosol–chemistry–climate model ECHAM6.3.0-HAM2.3-MOZ1.0. *Geosci. Model Dev.* **11**, 3833–3863 (2018).
- Lack, D. A. *et al.* Bias in Filter-Based Aerosol Light Absorption Measurements Due to Organic Aerosol Loading: Evidence from Ambient Measurements. *Aerosol Science and Technology* **42**, 1033–1041 (2008).
- Lewis, K., Arnott, W. P., Moosmüller, H. & Wold, C. E. Strong spectral variation of biomass smoke light absorption and single scattering albedo observed with a novel dual-wavelength photoacoustic instrument. *J. Geophys. Res.* **113**, D16203 (2008).
- Liu, S. *et al.* Aerosol single scattering albedo dependence on biomass combustion efficiency: Laboratory and field studies. *Geophysical Research Letters* **41**, 742–748 (2014).
- Martins, J. V. *et al.* Effects of black carbon content, particle size, and mixing on light absorption by aerosols from biomass burning in Brazil. *Journal of Geophysical Research: Atmospheres* **103**, 32041–32050 (1998).
- May, A. A. *et al.* Aerosol emissions from prescribed fires in the United States: A synthesis of laboratory and aircraft measurements: Aerosols from US prescribed fires. *J. Geophys. Res. Atmos.* **119**, 11,826–11,849 (2014).
- Onasch, T. B. *et al.* Single Scattering Albedo Monitor for Airborne Particulates. *Aerosol Science and Technology* **49**, 267–279 (2015).

Petzold, A. *et al.* Evaluation of Multiangle Absorption Photometry for Measuring Aerosol Light Absorption. *Aerosol Science and Technology* **39**, 40–51 (2005).

Pokhrel, R. P. *et al.* Parameterization of single-scattering albedo (SSA) and absorption Ångströmexponent (AAE) with EC / OC for aerosol emissions from biomass burning. *Atmos. Chem. Phys.* **16**, 9549–9561 (2016).

Saleh, R. From Measurements to Models: Toward Accurate Representation of Brown Carbon in Climate Calculations. *Curr Pollution Rep* (2020) doi:[10.1007/s40726-020-00139-3](https://doi.org/10.1007/s40726-020-00139-3).

Tiitta, P. *et al.* Chemical composition, main sources and temporal variability of PM₁ aerosols in southern African grassland. *Atmospheric Chemistry and Physics* **14**, 1909–1927 (2014).

Val Martin, M. *et al.* Smoke injection heights from fires in North America: analysis of 5 years of satellite observations. *Atmos. Chem. Phys.* **10**, 1491–1510 (2010).

Weingartner, E. *et al.* Absorption of light by soot particles: determination of the absorption coefficient by means of aethalometers. *Journal of Aerosol Science* **34**, 1445–1463 (2003).

WMO/GAW. Aerosol Measurement Procedures, Guidelines and Recommendation, Report No. 227 (World Meteorological Organization, Geneva, Switzerland, 2016).

Zhang, K. *et al.* The global aerosol-climate model ECHAM-HAM, version 2: sensitivity to improvements in process representations. *Atmos. Chem. Phys.* **12**, 8911–8949 (2012).

Reviewer 2

This paper argues that, as the title suggests, biomass burning aerosols in current climate models tend to lead to too much absorption of sunlight. The result is perhaps important to the climate modeling community, because the lingering uncertainty in model sensitivity to increasing greenhouse gas forcing is commingled with our uncertainty in the amount to which aerosols may be masking greenhouse warming by enhancing scattering of sunlight. It is the second of those uncertainties where this paper is arguing we have a systematic bias in models. However, the paper does not quite get to exactly the uncertainty in radiative forcing of biomass burning aerosols. The novel contribution of the paper is documenting a systematic bias in the intrinsic optical properties of biomass burning particles in models and helpfully points to some aspects of the parameterization of these intrinsic properties that might be adjusted to better match in-situ observations of these properties. The paper may be suitable for publication with some revisions as suggested below.

Reply: We thank the anonymous reviewer for their positive comments. We have reworded the abstract and other parts of the paper where we have used the word “uncertainty” to indicate “bias” in the optical properties of biomass burning (BB) aerosols in models.

The main evidence for aerosols that are too absorbing is a comparison of scatter plots of single-scattering albedo of aerosols against the ratio of black carbon mass to total carbon mass in field campaign observations and several models for locations dominated by biomass burning aerosols. They also show the mass absorption coefficient relative to the black carbon to total carbon ratio. Both indicate that the optical properties of the biomass burning particles is more absorbing in the model compared to observations. Curiously, the paper does not discuss quantitatively at all the net biomass burning radiative forcing of the models compared against the observations in figures 2 and 3.

Reply: Given that the AeroCom Phase III model data used in this study was originally compiled to address questions about BB emissions (amount and height) and BB distributions, emphasis was placed on aerosol emissions and concentrations in the model output. As a result, necessary TOA flux/DRE was not provided among these simulations. The exceptions are the ECHAM6.3-HAM2.3 and GEOS-Chem data. Following the reviewer’s comment, we added a comparison of CAM5.4 and these two models in a supplementary table (Supp. Table 13), and included the ECHAM6.3-HAM2.3 BB radiative effect (RE) and available literature values of model BB direct radiative effect (DRE) (i.e., RE due to aerosol-radiation interactions) in our concluding remarks (see our reply to the next comment).

Supplementary Table 13: Comparison of globally averaged BB radiative effect due to aerosol-radiation interactions (RE_{ari}) from CAM5.4, ECHAM6.3-HAM2.3, and GEOS-Chem simulations (Saleh et al., 2015). The CAM5.4 simulation RE_{ari} is calculated for specific years based on availability of GEOS-Chem and ECHAM6.3-HAM2.3 data.

Model Simulation	Simulation Year	BB RE _{ari} (W m ⁻²)
CAM5.4	2005	0.043
CAM5.4_BrC	2005	0.268
CAM5.4_BrCbl	2005	0.215
CAM5.4_ALL	2005	-0.040
GEOS-Chem	2005	
NA+EM		-0.460
A+EM		-0.240
NA+IM		-0.070
A+IM		0.050
CAM5.4	2008	0.042
CAM5.4_BrC	2008	0.269
CAM5.4_BrCbl	2008	0.200
CAM5.4_ALL	2008	-0.022
ECHAM6.3-HAM2.3	2008	0.082

They do note that the IPCC estimates net radiative forcing at 0.0 +/- 0.2 W m⁻². And they find that adjustments to aerosol parameterizations in one model lead to forcing ranging from 0.06 to -0.01 W m⁻², which is within the range of uncertainty of the IPCC estimate. I was unclear whether the implication of the paper is that the IPCC value is likely too positive and should be revised downward? Or whether the expected value of the IPCC estimate might be reasonable, but that the models under consideration in this paper have forcing values that are more positive? Or perhaps the models may get a net forcing value that is reasonable, but do so for the wrong reasons, since the intrinsic aerosol properties are not correct?

Reply: The intended message of the paper is that the IPCC value for BB aerosol radiative forcing is likely too high and should be revised downward. This is drawn from the majority of the analyzed models reporting SSA that are lower than observations, coupled with sensitivity tests in CAM5.4 that show improvement (increasing BB SSA) leading to a reduction in RE_{ari}. We note that the data from these AeroCom models have been used by the IPCC estimate. We have reworded our conclusion to make this clearer:

Line 509: “The models used in this study exhibit a wide range in BB RE_{ari}, from warming (0.059±0.009 W m⁻² (CAM5.4), 0.082 W m⁻² (ECHAM6.3-HAM2.3, Supplementary Table 13), 0.05 W m⁻² (GEOS-Chem) to cooling (-0.17±0.23 W m⁻² (HadGEM3) and -0.07 to -0.46 W m⁻² (GEOS-Chem)). However, most of these models exhibit overestimated BB absorptivity in Fig. 2. Changes to BB aerosol microphysical properties in CAM5.4 resulted in a change of BB RE_{ari} from positive to negative, with the most model improvement from changes to BB aerosol mixing state. The expectation is that improving overall model agreement with observations in Fig. 2 through similar changes will reduce simulated BB RE_{ari} across these models. This suggests that current estimates for BB RE_{ari} are biased high. Improvements to other models may lower overall model RE_{ari} in larger model ensemble datasets such as the sixth phase of the Coupled Model

Intercomparison Project (CMIP6)⁷⁰ or the IPCC AR5⁷¹. The changes to RE_{ari} of models will vary based on differences in microphysical treatments, and more work needs to be done to understand the responses in models to BB aerosol modifications. However, improvements to BB aerosols in models based on this SSA versus BC:TC framework have the potential to reduce bias in BB aerosol radiative forcing and establish BB aerosol as a cooling component in Earth's atmosphere.”

As correctly noted by the authors, there are a host of other factors that affect the net forcing that are not addressed here. The second half of the paper is dedicated to the sensitivity of the net radiative forcing in one model to the aerosol parameterizations, so the paper would be more impactful if the overall implications could be more specifically stated. In particular, this would help clarify whether this paper is merely showing that uncertainty in biomass burning aerosol properties is contributing to the wide uncertainty in biomass burning radiative forcing, or whether the models are truly overestimating aerosol absorption, as the title is boldly stating.

Reply: We thank the reviewer for pointing this out. We have made modifications to the paper to make more apparent, our message that models overestimate BB aerosol absorption. We also add statements that put the CAM5.4 radiative forcing tests in perspective when compared to the other models in the study and emphasize the importance of accurately treating aerosol mixing state in these results. We also reworded our concluding paragraph (see our reply to the previous comment). Modifications include:

Line 42: “Our findings suggest that current modeled BB contributes less to warming than previously thought, largely due to treatments of aerosol mixing state.”

Line 104: “Sensitivity tests are conducted with the NCAR Community Atmosphere Model version 5 (CAM5)⁴² to analyze the global radiative forcing consequences of improvements within this framework, with implications for the other models in this study.”

It is also very clear from looking at figures 2 and 3 that the models do a poor job of capturing the range and distribution of the ratio of black carbon mass to total carbon mass (BC:TC), which is attributed by the authors to the emissions, which are not adjusted in the experiments presented in the paper. While the sensitivity studies do indicate that CAM5 can be adjusted to yield a more realistic dependence of single-scattering albedo on the BC:TC ratio, and that doing so yields a more positive net radiative forcing of biomass burning aerosols. Presumably, one could achieve a similar result, at least qualitatively, on the net radiative forcing by adjusting the emission to reduce the BC fraction of the carbon emitted from biomass burning. Is there an indication of whether the magnitude of the result in net radiative forcing might be comparable from such an adjustment?

Reply: Following the reviewer's comment, we conducted sensitivity tests with model simulations for 2003-2005 with CAM6 (the latest version of CAM with the same aerosol

treatment as CAM5.4 we used; the earlier CAM versions are not supported anymore by NCAR), where we scaled BB BC emissions by 0.5 (while a different model than that used in the main text, this presents very similar results in the SSA vs BC:TC framework and in the patterns of radiative effect and AAOD). This sensitivity test results in similar BB radiative forcing changes ($\sim 0.07 \text{ W m}^{-2}$), but the effects in the SSA vs BC:TC framework are less promising in terms of model improvement compared to observations (see following figure).

We find that this change affects BC:TC quite dramatically, especially in regions with high BC:TC. It affects SSA to a lesser extent. The exception to this is lower SSA regions in the model with higher BC:TC, where a reduction in BC emissions would also result in a notable reduction in BB aerosol size (further lowering SSA from the default best-fit). If the aerosol size remained the same, we would expect the BCBB*0.5 simulation to follow the Default model best-fit. This is related to BC:TC being independent of concentration changes.

If the conclusion of the paper is that the specific choices for the parameterization of the size, mixing state and refractive index of biomass burning aerosols impacts the total absorption, is it possible that the sensitivity of the models to these details would be weaker if the BC:TC ratio were more similar to the observations?

Reply: As with the GFEDv3 BC:TC in CAM5.4, regions with higher BC:TC will be more sensitive to the changes proposed in this study due to high BC concentrations (greater effect due to BC RI change and absorption enhancement), and regions with lower BC:TC will be less sensitive to these changes due to lower BC content (less effect from BC RI change and absorption enhancement). Based on Fig. 2, in order to bring model BC:TC closer to observations in CAM5.4, BC:TC needs to be decreased at lower BC:TC and increased at higher BC:TC. This would mean that regions with higher BC:TC emissions (e.g., Africa) would be more sensitive to the microphysical changes, while regions with low BC:TC emissions (e.g., North America) would be less sensitive. This difference in response can be seen in the leveling of the regression line slopes in Fig. 4, moving from CAM5.4 to CAM_ALL. We expect a similar effect in other models.

There is little discussion of the uncertainties in the observations represented in figure 1 (and included in figures 2-4).

Reply: We included more discussion regarding instrument uncertainties in the Methods section. We also clarify that the error bars in Fig. 1-4 take into account these uncertainties in the main text.

Line 551: “The instruments used in this study have their own characteristic limitations that can lead to bias in SSA and BC:TC. Filter based measurements of absorption (PSAP, MAAP) and BC (MAAP) can be overestimated – especially in regions with high OA loadings – due to erroneous classification of multiple scattering as absorption (Bond et al., 1999; Lack et al., 2008; Petzold et al., 2005). Other non-linear responses can arise due to aerosol size distribution and increased filter loading (Weingartner et al., 2003). NEPH data is sensitive to aerosol size leading to a reduction in measured scattered light with a transition to forward and backward peaked scatter at sizes larger than ~300 nm (Anderson et al., 1998; Onasch et al., 2015). AMS and ACSM data can underreport OA concentrations due to issues with aerosol collection and particle ionization efficiency (Canagaratna et al., 2007; Tiitta et al., 2014). SP2 measurements tend to underestimate BC mass concentrations if not corrected for the high concentrations encountered in BB smoke plume measurements (May et al., 2014). This is in contrast to EC/OC analyzers, which can overestimate EC and underestimate OC due to misclassification of absorbing OC as EC (Saleh, 2020). PASS-3 and UWPAS exhibit small error in absorption coefficient arising from instrument dependent uncertainties (i.e., resonant frequency, resonator quality factor, microphone pressure, and laser power) (Lewis et al., 2008). Lastly, SSA calculated from UWPAS utilizes extinction coefficients calculated from the CAPS PMSSA (Foster et al., 2019), which suffers from a similar scattering bias at larger aerosol sizes as the NEPH.

Uncertainty estimates based on the various instrument adjustments, calibrations, corrections, and assumptions for all observations were added to the scatter plot data.”

There are citations in the supplementary tables, however, it would be good to include a statement in the body of the paper explaining what these error bars represent. Are these just variability in the measurements, instrument uncertainties, or combinations of these uncertainties?

Reply: These error bars take into account the instrument uncertainties. We added a statement to this effect in the revised manuscript.

Line 120: “Instrument uncertainty estimates for these data are represented by error bars in Fig. 1 (see Methods).”

The changes in net radiative forcing in the altered versions of the CAM5 model discussed in the paragraph beginning line 351 will depend on the albedo of the surface and/or clouds beneath the biomass burning aerosol. Are there responses in the clouds due to the

difference in the aerosol properties between these simulations that are likely to have a meaningful impact on the brightness of the surface beneath the biomass burning aerosol?

Reply: We check the effect of CAM5.4 simulations on low clouds, and the effect of clouds on direct radiative forcing (REari), and determine that the effects of cloud changes on REari are small. The following figure shows the effect of the different simulations on low cloud fraction (LCF; %), which could have an influence on the calculation of REari. This influence comes from changes in reflected light by the cloud with changing cloud properties, affecting changes in absorption of reflected light within the above-cloud aerosol layer (Sakaeda et al., 2011; Lu et al., 2018).

Hatching indicates significance in the ensemble year change to the 0.05 level.

From the previous figure we can see that cloud fraction does change with the different modifications, and intermittent changes in LCF correlate with REari (Fig. 5). The main regions where significant LCF changes correspond with significant changes in REari are

in regions of aerosol transport such as off the SE African coast, off the west coast of North America, and over the Arctic Ocean. In the case of CAM_Dg160 and CAM_ALL, significant changes in LCF in the Arctic are more pronounced and correspond to significant changes in RE_{ari} in the same region.

While these changes in LCF may be affecting RE_{ari}, we mention in the manuscript that aerosol size changes in these models are also contributing to changes in RE_{ari}:

Line 303: “A more likely scenario for these changes in the Arctic is enhanced dry and wet deposition due to increased aerosol size. This agrees with a decrease in Mode 4 POA and BC lifetimes in the Arctic from CAM5.4 (3.06 days for POA, 3.09 days for BC) to CAM_DG160 (2.37 days for POA, 2.39 days for BC) (Supplementary Table 12).”

We acknowledge aerosol size and cloud effects in our discussion:

Line 385: “In this study, the external mixing for aerosol optical properties with the default internally mixed treatment of aerosol hygroscopicity minimizes these effects in CAM_EMIX, but the effects on cloud properties and aerosol lifetime in the model from changing aerosol size cannot be avoided in CAM_DG160 and CAM_ALL. These changes have the potential to affect RE_{aci}, which is an important consideration when calculating total RE or RF.”

The statement in lines 447-448 that “assuming internal mixtures for aerosols in models has been considered a more realistic approach” needs a reference. That this particular factor seems to be impactful on the relationship of SSA to BC:TC seems to be an important result from this paper.

Reply: Following the reviewer’s comment, we have reworded this statement and added references:

Line 361: “Assuming that model aerosol is internally mixed is intended to represent the observed mixing states of these particles (Adachi et al., 2010; Liu et al., 2015; Martins et al., 1998). However, these idealized uniformly mixed (well-mixed RI) and perfectly coated (core-shell) aerosol treatments in models overestimate aerosol absorption enhancement (Cappa et al., 2012).”

References

Adachi, K., Chung, S. H. & Buseck, P. R. Shapes of soot aerosol particles and implications for their effects on climate. *Journal of Geophysical Research: Atmospheres* **115**, (2010).

Anderson, T. L. & Ogren, J. A. Determining Aerosol Radiative Properties Using the TSI 3563 Integrating Nephelometer. *Aerosol Science and Technology* **29**, 57–69 (1998).

Bond, T. C., Anderson, T. L. & Campbell, D. Calibration and Intercomparison of Filter-Based Measurements of Visible Light Absorption by Aerosols. *Aerosol Science and Technology* **30**, 582–600 (1999).

Brown, H. *et al.* Radiative effect and climate impacts of brown carbon with the Community Atmosphere Model (CAM5). *Atmospheric Chemistry and Physics* **18**, 17745–17768 (2018).

Canagaratna, M. R. *et al.* Chemical and microphysical characterization of ambient aerosols with the aerodyne aerosol mass spectrometer. *Mass Spectrom. Rev.* **26**, 185–222 (2007).

Cappa, C. D. *et al.* Radiative Absorption Enhancements Due to the Mixing State of Atmospheric Black Carbon. *Science* **337**, 1078–1081 (2012).

Foster, K., Pokhrel, R., Burkhart, M. & Murphy, S. A novel approach to calibrating a photoacoustic absorption spectrometer using polydisperse absorbing aerosol. *Atmos. Meas. Tech.* **12**, 3351–3363 (2019).

Lack, D. A. *et al.* Bias in Filter-Based Aerosol Light Absorption Measurements Due to Organic Aerosol Loading: Evidence from Ambient Measurements. *Aerosol Science and Technology* **42**, 1033–1041 (2008).

Lewis, K., Arnott, W. P., Moosmüller, H. & Wold, C. E. Strong spectral variation of biomass smoke light absorption and single scattering albedo observed with a novel dual-wavelength photoacoustic instrument. *J. Geophys. Res.* **113**, D16203 (2008).

Liu, S. *et al.* Enhanced light absorption by mixed source black and brown carbon particles in UK winter. *Nature Communications* **6**, (2015).

Lu, Z. *et al.* Biomass smoke from southern Africa can significantly enhance the brightness of stratocumulus over the southeastern Atlantic Ocean. *Proceedings of the National Academy of Sciences* **115**, 2924–2929 (2018).

Martins, J. V. *et al.* Effects of black carbon content, particle size, and mixing on light absorption by aerosols from biomass burning in Brazil. *Journal of Geophysical Research: Atmospheres* **103**, 32041–32050 (1998).

May, A. A. *et al.* Aerosol emissions from prescribed fires in the United States: A synthesis of laboratory and aircraft measurements: Aerosols from US prescribed fires. *J. Geophys. Res. Atmos.* **119**, 11,826–11,849 (2014).

Onasch, T. B. *et al.* Single Scattering Albedo Monitor for Airborne Particulates. *Aerosol Science and Technology* **49**, 267–279 (2015).

Petzold, A. *et al.* Evaluation of Multiangle Absorption Photometry for Measuring Aerosol Light Absorption. *Aerosol Science and Technology* **39**, 40–51 (2005).

Sakaeda, N., Wood, R. & Rasch, P. J. Direct and semidirect aerosol effects of southern African biomass burning aerosol. *Journal of Geophysical Research* **116**, (2011).

Saleh, R. From Measurements to Models: Toward Accurate Representation of Brown Carbon in Climate Calculations. *Curr Pollution Rep* (2020) doi:[10.1007/s40726-020-00139-3](https://doi.org/10.1007/s40726-020-00139-3).

Tiitta, P. *et al.* Chemical composition, main sources and temporal variability of PM₁ aerosols in southern African grassland. *Atmospheric Chemistry and Physics* **14**, 1909–1927 (2014).

Weingartner, E. *et al.* Absorption of light by soot particles: determination of the absorption coefficient by means of aethalometers. *Journal of Aerosol Science* **34**, 1445–1463 (2003).

Reviewer 3

The authors use simulations from the phase III of the aerocom model intercomparison exercise and compare them to a range of observational data to derive the conclusion that the biomass-burning aerosol optical properties in atmospheric composition models are generally too absorbing. They focus on CAM5, which they use to test a range of proposed changes in the BB aerosol refractive index and assumed size distribution in order to reduce this bias. The manuscript is very well written and presented, and the authors master their subject very well. An impressive range of observational datasets are used in a very synthetic way (by using scatterplots of SSA vs BC/TC ratio) to assess simulated optical properties; the authors have been careful to restrict this observational dataset to their species of interest. The fact that several well known models are evaluated add credence to their claim about the BB aerosol properties being too absorbing. Using SSA vs BC/TC ratio is a pertinent metric as it allows the authors to focus exclusively on the aerosol optical properties and not on the fact that BB emissions are over or underestimated. The proposed small scale changes in the BB aerosol optical properties are easy to implement and very interesting to the atmospheric composition modelling community. I recommend the manuscript to be accepted and have only minor questions and remarks:

Reply: We thank the reviewer for the positive and encouraging comments on our work.

- How does the work on the lead author on Brown Carbon (Brown et al., 2018, ACP) fit in this study? This is mentioned in passing but I think it deserves a more developed discussion. In this work, brown carbon is apparently not used, is there a reason for this?

Reply: Following the reviewer's comment, we have added the simulations from Brown et al. (2018) to Figures 2 (panel b) and 3 (panel b), analyses of brown carbon simulations at different wavelengths (see our reply to the next comment), and discussion regarding brown carbon to the main text.

Figure 2: Comparison of the observation data from Fig. 1a (in gray) to a variety of global climate models. The best-fit to the model data is represented by the solid blue line. The model data is from (a) this study, (b) Brown et al.¹⁷, (c-g) AeroCom Phase-III simulations, and (f) Saleh et al. (2015). They are (a) CAM5.4, (b) CAM5.4 (w/ BrC), (c) CAM5.3, (d) ECHAM6.3-SALSA2.0, (e) ECHAM6.3-HAM2.3, (f) HadGEM3, (g) OsloCTM2, and (h) GEOS-Chem. Model data is representative of BB influenced regions which are color coded in the model output and are specified on the global plot in panel (i). Observational data from the specific regions are averaged, and the regional average and range of the BC:TC observational data is included at the bottom of the plot (color coded squares). The 2 CAM5.4 simulations with BrC (b) represent brown carbon with and without photochemical bleaching (BrC and BrCbl, respectively). The four GEOS-Chem simulations (f) represent the four model simulations from Saleh et al. (2015): non-absorbing organics and externally mixed aerosols (NA+EM); absorbing organics and externally mixed aerosols (A+EM); non-absorbing organics and internally mixed aerosols (NA+IM); and absorbing organics and internally mixed aerosols (A+IM). Given that the CAM5.4 simulations were run for a number of years, and the AeroCom and GEOS-Chem simulations are for a single year, interannual variation in model slope and intercept is represented by dashed blue lines in panels a and b (long dash = BrC and short dash = BrCbl in panel b).

Figure 3: BB mass absorption cross-section (MAC_{BB} ; abs. coeff. / $([BC] + [OA])$, $m^2 g^{-1}$) versus BC:TC ratio. Available MAC_{BB} observations are plotted from Fig. 1. The models are the same as in Fig. 2. The solid gray line is the best fit to observations, with dashed lines representing the 95% confidence intervals of the fit. The solid blue line is a best fit to the model data. Thick black lines constrain a reasonable range in BB MAC ¹⁸.

We added the following discussion in the main text (Line 212): “What this collection of models shows is that all of the models overestimate BB aerosol absorption relative to extinction to some extent. Some models perform better than others in this comparison, especially with smaller BB aerosol imaginary refractive indices and external mixing assumptions. Of the models included in this comparison, two CAM5.4 simulations and two GEOS-Chem simulations include the effects of BrC on aerosol absorption. CAM5.4 simulations with BrC from Brown et al.¹⁷ exacerbate the excessive absorption in the

model, with photochemical bleaching effects (CAM5.4_BrCbl) slightly improving model-observation slope agreement. At shorter wavelengths (400 nm) where the effects of BrC absorption are greater, CAM5.4 agrees better with observations (Supplementary Fig. 16) than CAM5.4_BrC (Supplementary Fig. 17), which continues to exhibit lower SSA than observations. This comparison emphasizes that CAM5.4 SSA is too low at both 400 and 550 nm wavelengths, and including BrC in the model further reduces model SSA. In order to help understand the underlying causes of the BB absorption enhancement in the models, the following section will address areas of improvement in CAM5.4, and how these improvements affect BB SSA.”

- This is linked with the previous question: the manuscript focuses on the absorption in the visible part of the spectrum (550nm). What about other wavelengths, and in particular the near UV where brown carbon could make a difference?

Reply: Following the reviewer’s comment, we have included comparisons for CAM5.4 and CAM5.4 (with brown carbon) at the wavelengths 400 nm, 550 nm, and 700 nm. These comparisons are plotted in Supplementary Figure 16.

Supplementary Figure 16: Same comparison as Fig. 2a, but comparing SSA versus BC:TC at three different wavelengths: (a) 400 nm, (b) 550 nm, and (c) 700 nm. Observations for panels (b) and (c) are the same as for Fig. 1, while observations for panel (a) are from Pokhrel et al. (2016), which are part of observations in b) and c).

Supplementary Figure 17: Same as Supplementary Figure 16, but using CAM5.4_BrC.

Supplementary Figure 16 shows that CAM5.4 underestimates SSA at 550 and 700 nm, but agrees better with observations at 400 nm wavelength. This is surprising given that these wavelengths are more affected by brown carbon, which CAM5.4 does not include. When brown carbon is included in this comparison (Supplementary Figure 17), the model shows reduced SSA at all wavelengths compared to Supplementary Figure 16.

- This work is of course of interest to climate model (and the extension about radiative forcing is of special interest there), but also to general atmospheric composition modellers. This could be mentioned, either in the title or in the abstract.

Reply: We thank the reviewer for their comment. Due to space limitations, we only changed the first sentence of the abstract to include an aerosol composition statement. Further comments regarding composition modeling were added in our Discussion/Conclusion

Line 33: “Uncertainty in the representation of biomass burning (BB) aerosol composition and optical properties in climate models contributes to a range in modeled aerosol effects on incoming solar radiation.”

Line 380: “These findings indicate a need for physical model treatments of aerosol composition that reduce absorption enhancement while representing hygroscopicity changes of aged, internally mixed aerosols.”

- What is the impact of the proposed changes on extinction? This is mentioned I think in the discussion about the impact on radiative properties, but I think a Figure showing global AOD and AAOD as simulated by CAMS5 with and without the proposed changes in aerosol optical properties would be very interesting to the reader.

Reply: Following the reviewer’s comment, we included a figure in the supplementary showing BB AOD for CAM5.4 and the sensitivity experiments (similar to AAOD in the main text, Figure 6):

Supplementary Figure 19: Same as Fig. 6 but for BB aerosol optical depth (AOD).

There is a slight decrease in global BB AOD with increasing aerosol size. Interestingly, the pattern of these changes increases the AOD of BB aerosol over the source regions, especially Africa, and lowers AOD for transported BB aerosol (Supp. Fig. 19e), mainly due to changes in BB size (CAM_Dg160). Decrease in BB AOD over the Saharan Desert and Yemen/Oman is likely due to enhanced deposition/reduced transport of BB aerosols with an increase in aerosol size, while increases over African source regions can be explained by enhanced scattering with increasing size.

- While the BB emissions in themselves are indeed out of the scope of this paper, it would be nice to have possibly some direct comparison of simulated values against AERONET for example (for AAOD and AOD if relevant), to check if the proposed changes do have a positive impact against AERONET. Of course, this evaluation would be impacted by possible defaults (and underestimation) of GFEDv2, which the authors mentioned already.

Reply: We thank the reviewer for this suggestion. Following the reviewer’s comment, we show comparisons of the CAM5.4 simulations of SSA, AOD, and AAOD at 675 nm against observations at AERONET sites used in Brown et al. (2018) dominated by BB aerosols. Comparisons of model and AERONET SSA show that CAM5.4 tends to underestimate SSA in BB regions, and the modifications shown in Fig. 4 improve model agreement with AERONET observations. Comparisons of model and AERONET AAOD show that the default model tends to underestimate AAOD compared to observations. There is an increase in AAOD with increasing aerosol size (as in Fig. 6), which improves the model-AERONET agreement in Africa and some South American regions. In other cases AAOD seems to be reduced more than observations, suggesting that some combination of the CAM sensitivity tests and brown carbon might be necessary.

Supplementary Figure 20: AERONET and model comparison of single scatter albedo (SSA) for the wavelengths of 675 nm for AERONET and 700 nm for CAM5.4. This comparison is the same as that in Brown et al. (2018), and compares AERONET sites influenced by African (a-c), South American (d-f), and Arctic (g-i) BB emissions. Values below the upper x-axis indicate percentage of available data in the 9-year period.

Supplementary Figure 21: Same as Supplementary Figure 20 but for AERONET comparison of aerosol absorption optical depth (AAOD) (at wavelengths of 675 nm for AERONET and 700 nm for CAM5.4).

Comparisons of AERONET and model AOD show that the model simulations also underestimate AOD – driving the underestimation in AAOD. The enhanced scattering with increased BB aerosol sizes improves model-observation agreement in all of the AERONET BB sites. As the reviewer points out, the disagreement between AERONET and the models may be partly due to limitations in the GFEDv3 dataset.

Supplementary Figure 22: Same as Supplementary Figure 21 but for AERONET comparison of aerosol optical depth (AOD) (at wavelengths of 675 nm for AERONET and 700 nm for CAM5.4).

We acknowledge these figures in the revised manuscript.

Line 265: “Model SSA improvement is also noted in comparisons with AERONET BB sites (Supplementary Figure 20).”

Line 329: “Comparisons to AERONET observations show that CAM5.4 tends to underestimate AOD and AAOD in BB regions. Increasing aerosol size (CAM_Dg160) improves the AOD and AAOD model performance in African BB regions (Supplementary Figures 21, 22), though the tendency of GFEDv3 to underestimate BB emissions²⁵ may also explain this disagreement.”

References

Brown, H. *et al.* Radiative effect and climate impacts of brown carbon with the Community Atmosphere Model (CAM5). *Atmospheric Chemistry and Physics* 18, 17745–17768 (2018).

Pokhrel, R. P. *et al.* Parameterization of Single Scattering Albedo (SSA) and Absorption Angstrom Exponent (AAE) with EC/OC for Aerosol Emissions from Biomass Burning. *Atmospheric Chemistry and Physics Discussions* 1–27 (2016).

REVIEWERS' COMMENTS

Reviewer #2 (Remarks to the Author):

The fundamental struggle that I have had with this paper is that the authors' primary assertion, that climate warming from biomass burning aerosols in current climate models is overestimated, cannot really be directly assessed because there is no estimate of the true radiative effect of biomass burning aerosols that is independent of the models themselves. So, the authors have evaluated just one critical element of the models, the specification of the optical properties of the biomass burning aerosols. Indeed, they find that the biomass burning aerosol optical properties in the models, at least on a per-particle basis, do appear to be too absorbing when compared to available observations of these same aerosol optical properties. The implication then is that these aerosols, in the aggregate, contribute "less to warming than previously thought" (line 43). But this conclusion is ultimately only implied by their comparisons of models to observations and is not directly demonstrated.

Reviewer 1 expressed a similar sentiment when pointing out that the variations in net radiative effect of biomass burning aerosols among the models is significantly larger (0.23 W m^{-2} [lines 422-423 in the revised manuscript], or 0.40 W m^{-2} quoted by the reviewer from the IPCC) than the change attributable to the adjustments the authors are recommending to the aerosol optical properties, -0.07 W m^{-2} . Thus, while the authors may be correct in arguing that the optical properties of the aerosols in many of these models appear to be similarly biased toward greater absorption for a given ratio of black carbon to total carbon when compared to available observations, and adjusting all of those models in the same direction to counter that bias would indeed yield less climate warming when averaged over all of the models, there appear to be many remaining uncertainties in the model treatment of biomass burning aerosols that are not addressed in this paper and lead to lingering widespread uncertainties. Left unexplored are other potential adjustments to aspects such as emissions rates, deposition rates and transport of biomass burning aerosols that if similarly adjusted might lead to changes in the biomass burning radiative effect of similar or greater magnitude to the adjustments to the optical properties they are recommending.

Nevertheless, I find the results documenting the biases in the simulated single-scattering albedo and mass absorption cross-section relative to observations to be compelling and important for consideration within the aerosol/climate modeling community. In my view the authors have responded to the technical issues raised by all the reviewers adequately. Indeed, the authors do a commendable job of providing a variety of analyses that corroborate their main observational result.

In a few places, I still feel that the authors have used language that overstates the level of certainty in the implied result of less warming. For example, in lines 413-415, the authors state that "values of AAOD in Bond et al. are overestimated when compared to Kinne due to overestimation of the anthropogenic BC contributions and an underestimation of mineral dust contributions." Here the authors appear to be favoring the result of the Kinne et al. paper, which is a modeling study that is presumably better aligned with their assertion that biomass burning aerosol contribute less to warming than previously thought. As noted above, such global radiative effects of aerosols are not independent of the models. So, a more balanced argument would be that the values of AAOD in Bond et al. are larger than those in the Kinne paper. An updated estimate of black carbon aerosol effects from a global modeling study such as Kinne's might imply stronger contributions from anthropogenic BC and weaker contributions from mineral dust in the preceding work. A proper comparison of the model with independent observations would be required to demonstrate an "overestimation" of the former and an "underestimation" of the later.

Similarly, in lines 425-426 the authors argue that "changes in BB aerosol microphysical properties in CAM5.4 resulted in a change of BB RE_ari from positive to negative, with the most model improvement from changes to BB aerosol mixing state." This wording implies that altering the model physics such that the biomass burning aerosol radiative effect in the model from positive to negative is

an improvement. But the notion that the decrease in the net radiative effect of biomass burning aerosols constitutes an "improvement" is an inference from this study and not directly demonstrated, as noted above. A more accurate portrayal would be to argue that altering the model physics such that the aerosol optical properties better agree with observations changes the net radiative effect from positive to negative.

I feel the paper is a valuable contribution and suitable for publication. Better attention to the word choices in some places where the implications for global aerosol effects are addressed would better convey how these results relate to the state of understanding of biomass burning aerosol effects.

Reviewer #3 (Remarks to the Author):

Dear authors,

Thanks for your answers to my comments; the proposed changes of the manuscript are fine by me.

We thank the editor and the anonymous reviewer for the encouraging comments and constructive suggestions on the manuscript. Below, we explain how the comments and suggestions for the second round of revisions are addressed and make note of the revisions in the revised manuscript. The Reviewer comments are in **blue**, our replies are in **black**, and the corresponding changes to the manuscript are in **red**.

Reviewer 2

The fundamental struggle that I have had with this paper is that the authors' primary assertion, that climate warming from biomass burning aerosols in current climate models is overestimated, cannot really be directly assessed because there is no estimate of the true radiative effect of biomass burning aerosols that is independent of the models themselves. So, the authors have evaluated just one critical element of the models, the specification of the optical properties of the biomass burning aerosols. Indeed, they find that the biomass burning aerosol optical properties in the models, at least on a per-particle basis, do appear to be too absorbing when compared to available observations of these same aerosol optical properties. The implication then is that these aerosols, in the aggregate, contribute "less to warming than previously thought" (line 43). But this conclusion is ultimately only implied by their comparisons of models to observations and is not directly demonstrated.

Reviewer 1 expressed a similar sentiment when pointing out that the variations in net radiative effect of biomass burning aerosols among the models is significantly larger (0.23 W m^{-2} [lines 422-423 in the revised manuscript], or 0.40 W m^{-2} quoted by the reviewer from the IPCC) than the change attributable to the adjustments the authors are recommending to the aerosol optical properties, -0.07 W m^{-2} . Thus, while the authors may be correct in arguing that the optical properties of the aerosols in many of these models appear to be similarly biased toward greater absorption for a given ratio of black carbon to total carbon when compared to available observations, and adjusting all of those models in the same direction to counter that bias would indeed yield less climate warming when averaged over all of the models, there appear to be many remaining uncertainties in the model treatment of biomass burning aerosols that are not addressed in this paper and lead to lingering wide-spread uncertainties. Left unexplored are other potential adjustments to aspects such as emissions rates, deposition rates and transport of biomass burning aerosols that if similarly adjusted might lead to changes in the biomass burning radiative effect of similar or greater magnitude to the adjustments to the optical properties they are recommending.

Reply: We appreciate this critique of our work. We feel that addressing all of these concerns is out of the scope of this study, but we do not intend to ignore other potential uncertainties in BB radiative effects. We already mention the potential impact of different mass emissions in our discussion (Lines 387-402), but don't specifically mention the impacts of transport and deposition on direct and indirect radiative effects. We include a statement in our discussion to address your comment above.

(Line 383): “These changes have the potential to affect RE_{ari} and the RE due to aerosol-cloud interactions by altering aerosol transport and deposition. Addressing uncertainty in these processes is also important, possibly leading to changes in BB RE_{ari} of similar or greater magnitude than the optical property changes detailed in this study.”

Nevertheless, I find the results documenting the biases in the simulated single-scattering albedo and mass absorption cross-section relative to observations to be compelling and important for consideration within the aerosol/climate modeling community. In my view the authors have responded to the technical issues raised by all the reviewers adequately. Indeed, the authors do a commendable job of providing a variety of analyses that corroborate their main observational result.

In a few places, I still feel that the authors have used language that overstates the level of certainty in the implied result of less warming. For example, in lines 413-415, the authors state that “values of AAOD in Bond et al. are overestimated when compared to Kinne due to overestimation of the anthropogenic BC contributions and an underestimation of mineral dust contributions.” Here the authors appear to be favoring the result of the Kinne et al. paper, which is a modeling study that is presumably better aligned with their assertion that biomass burning aerosol contribute less to warming than previously thought. As noted above, such global radiative effects of aerosols are not independent of the models. So, a more balanced argument would be that the values of AAOD in Bond et al. are larger than those in the Kinne paper. An updated estimate of black carbon aerosol effects from a global modeling study such as Kinne’s might imply stronger contributions from anthropogenic BC and weaker contributions from mineral dust in the preceding work. A proper comparison of the model with independent observations would be required to demonstrate an “overestimation” of the former and an “underestimation” of the later.

Reply: We appreciate the reviewers comment. We have changed the wording to reflect a more balanced reporting of the data from the literature.

(Line 409): “Additionally, the values of AAOD in Bond et al.¹ are larger than those in Kinne⁶⁹ due to stronger anthropogenic BC contributions and weaker mineral dust contributions in the former study.”

Similarly, in lines 425-426 the authors argue that “changes in BB aerosol microphysical properties in CAM5.4 resulted in a change of BB RE_{ari} from positive to negative, with the most model improvement from changes to BB aerosol mixing state.” This wording implies that altering the model physics such that the biomass burning aerosol radiative effect in the model from positive to negative is an improvement. But the notion that the decrease in the net radiative effect of biomass burning aerosols constitutes an “improvement” is an inference from this study and not directly demonstrated, as noted above. A more accurate portrayal would be to argue that altering the model physics such

that the aerosol optical properties better agree with observations changes the net radiative effect from positive to negative.

Reply: We see how this statement may misrepresent our results. We have changed it following the above recommendation.

(Line 420): “Altering the CAM5.4 BB aerosol microphysical properties to better agree with observations resulted in a change of BB RE_{ari} from positive to negative, with the most improvement in model SSA resulting from changes to BB aerosol mixing state.”

I feel the paper is a valuable contribution and suitable for publication. Better attention to the word choices in some places where the implications for global aerosol effects are addressed would better convey how these results relate to the state of understanding of biomass burning aerosol effects.

Reply: Many thanks to the anonymous reviewer for their time and additional recommendations.